# One Bias After Another: Mechanistic Reward Shaping and Persistent Biases in Language Reward Models

Daniel Fein [* 1]   Max Lamparth [* 1]   Violet Xiang [1]   Mykel J. Kochenderfer [1]   Nick Haber [1]

## Abstract

Reward Models (RMs) are crucial for online alignment of language models (LMs) with human preferences. However, RM-based preference-tuning is vulnerable to *reward hacking*, whereby LM policies learn undesirable behaviors from flawed RMs. By systematically measuring biases in five high-quality RMs, including the state-of-the-art, we find that issues persist despite prior work with respect to length, sycophancy, and overconfidence. We also discover new issues related to bias toward model-specific "styles" and answer-order. We categorize RM failures as tractable or resistant to linear intervention and propose a simple post-hoc intervention to mitigate low-complexity biases that arise from spurious correlations. Our proposed **mechanistic reward shaping** reduces targeted biases without degrading reward quality and while using minimal labeled data. The method is extensible to new biases, model-internal, and generalizes out-of-distribution.

## 1. Introduction

Reinforcement Learning from Human Feedback (RLHF) remains a popular method to control the behavior of language models (Ouyang et al., 2022; Wang et al., 2023; Lambert, 2025). However, RLHF is vulnerable to *reward hacking*, whereby optimizing an imperfect proxy reward function leads to poor performance according to a true reward function (Skalse et al., 2022; Lambert et al., 2023; Casper et al., 2023). Reward hacking during RLHF is a potential source of observed biases in language model behavior including length bias (Sanz-Guerrero & von der Wense, 2025; Singhal et al., 2024), position bias (Zhao et al., 2024b), overconfidence (Chen et al., 2023), and sycophantic user agreement (Sharma et al., 2024; Wen et al., 2025).

Though a significant body of work has addressed the susceptibility of RLHF to reward hacking, much prior work frames reward hacking as arising from reward misgeneralization based on linear spurious correlation, with trained policies learning to exploit flawed reward functions (Shuieh et al., 2025; Ng et al., 2025; Chen et al., 2024). However, this perspective diminishes the role of reward hacking resultant from the non-linear artifacts caused by reward misspecification. Here, we empirically distinguish biases amenable to lightweight linear correction (low-complexity) from those that resist it (high-complexity), providing a practical framework for prioritizing deployable interventions.

Motivated by the Linear Representation Hypothesis (Park et al., 2024a), we treat linear probe null-space projection as a practical first-order intervention that does not need to fully resolve a bias to be deployable, but rather serves as a tractable correction for biases whose dominant signal is approximately linear. We find that isolating and removing such directions via a null-space projection mitigates length-, uncertainty-, and position-related biases. In contrast, other RM failures appear more complex and are not well-approximated by a single linear direction, limiting the effectiveness of linear probe interventions. Addressing biases at the RM level is particularly impactful as RMs serve as the foundation for multiple alignment techniques beyond RLHF, including automated red-teaming, best-of-N sampling, and iterative data filtering. Moreover, model-internal interventions on RMs enable pointwise reward correction without requiring access to or modification of the policy optimization procedure, making them broadly applicable across deployment contexts.

Through systematic evaluation of five RMs (including state-of-the-art) across multiple dimensions, we identify both persistent and novel failure modes, introduce a data-efficient probe-nulling technique for tractable biases, and categorize biases as linear or complex, characterizing the limitations of linear interventions for complex artifacts. We make five core contributions:

- We demonstrate that previously discovered biases persist in state-of-the-art RMs, using labeled datasets to

[*]Equal contribution   [1]Stanford University.  Correspondence to: Daniel Fein <drfein@stanford.edu>, Max Lamparth <lamparth@stanford.edu>.

*Proceedings of the $43^{rd}$ International Conference on Machine Learning*, Seoul, South Korea. PMLR 306, 2026. Copyright 2026 by the author(s).

detect biases based on length, overconfidence, and sycophancy across five contemporary RMs.

- We identify two novel biases in RMs: (1) **position bias** (both in free-form text and multiple-choice settings), and (2) **model-style reward sensitivity**, where RMs systematically reward or penalize completions based on their distributional similarity to specific language model writing styles.

- We introduce an empirical taxonomy of low/high-complexity RM biases, distinguishing those where a linear first-order correction is deployable and sufficient (length, uncertainty, position) from those that resist simple linear intervention and require more fundamental solutions (sycophancy, model-style sensitivity).

- We introduce a **mechanistic reward shaping** approach using linear activation probes to mitigate low-complexity biases. This method is data-efficient, model-internal (enabling use beyond regular RLHF), and significantly reduces targeted biases.

- We show that the proposed intervention does not significantly degrade accuracy on RewardBench2, and that probes constructed from synthetic datasets generalize out-of-distribution.

We provide extensive empirical analysis across four diverse benchmarks (PlausibleQA, BigBench, GSM8K-MC, MMLU), demonstrating both the effectiveness of our intervention for tractable biases and the limitations of linear interventions for complex artifacts. The code and generated data are publicly available on GitHub[1] (MIT license).

## 2. Related Work

A significant body of work addresses **reward hacking in RLHF**. Existing methods toward this problem span data augmentation techniques (Kumar et al., 2025; Liu et al., 2025b), preference data analysis (Movva et al., 2026), RM token-level ranking analysis (Christian et al., 2025), robust RM training methods (Miao et al., 2024; Chen et al., 2024; Wang et al., 2025; Ng et al., 2025), and post-hoc reward shaping (Fu et al., 2025; Huang et al., 2025; Eisenstein et al., 2024). Some work found RMs rank completions from their own base-model lineage slightly higher on average using an inverse-rank analysis (Malik et al., 2026), but not in a continuous, model-conditioned style sensitivity. In contrast, we identify two previously uncharacterized RM biases and intervene directly on RM representations, enabling targeted removal of specific spurious features without retraining, while explicitly distinguishing low-complexity correlational biases from entangled, context-dependent artifacts.

---

[1] github.com/drfein/OneBiasAfterAnother

Post-hoc methods avoid costly retraining but are limited in generality. Prior length-bias corrections rely on explicit modeling of the length–reward relationship (Park et al., 2024b; Huang et al., 2025; Zhao et al., 2025), e.g., with length penalties. These methods debias under their target diagnostic, which is typically a global, prompt-pooled statistic, so they need not transfer to prompt-conditioned selection settings such as best-of-N (see Section 4.4). Our approach instead removes length-correlated directions in the RM latent space, enabling correction even when the functional form of the bias is unknown. Other post-hoc methods, such as reward shaping via bounded transformations (Fu et al., 2025) or RM ensembles (Eisenstein et al., 2024), regularize reward behavior globally but cannot isolate and selectively remove individual spurious features.

**Linear Probes:** The linear representation hypothesis posits that high-level concepts are represented linearly as directions in some representation space (Park et al., 2024a). It has proven instrumentally useful within the subfield of interpretability and Wu et al. (2025) show that difference-of-mean probes (which we adopt) constructed from language model activations can reliably detect concepts. Significant work has shown how these concept vectors can be used to steer generation behavior of language models (Turner et al., 2024; Lamparth & Reuel, 2024; Rimsky et al., 2024; Siu et al., 2026; Chen et al., 2025). A smaller body of work attempts to use the complement of a known activation subspace to steer model behavior. Ravfogel et al. (2020) null subspaces from an early tweet-classification model to remove unwanted biases. Casademunt et al. (2025) null out directions obtained through sparse autoencoders to steer language-model finetuning away from misaligned behavior.

Compared to prior debiasing methods, our intervention is lightweight and easily extensible to new biases for already-trained RMs, enabling reward correction without modifying downstream policy optimization algorithms. More broadly, existing RM debiasing and probing approaches do not distinguish between biases that are well-approximated by simple linear features and those arising from complex, entangled representations.

## 3. Background

### 3.1. Reward Bias Complexity

We categorize RM biases into two complexity classes based on their tractability under model-internal causal intervention. We use "mechanistic" in the narrow sense discussed by Saphra & Wiegreffe (2024), where interventions that identify and remove specific directions in activation space produce measurable causal changes in downstream reward behavior, rather than circuit-level explanations. Our taxonomy is therefore empirically descriptive and characterizes

which biases yield to a linear first-order fix, not whether they are strictly linear in a representational geometry sense:

*Low-complexity biases*, which correspond to relatively simple features (e.g., length, linguistic markers, positional information) that can be captured by linear directions in the model's representation space, and *high-complexity biases*, which arise from entangled, context-dependent factors resistant to simple linear decomposition (e.g., abstract concepts like sycophancy, model-style sensitivity, or fairness). We hypothesize that low-complexity biases can be mitigated by removing corresponding linear activation directions in latent space, while high-complexity biases require more sophisticated interventions. High-complexity biases plausibly resist linear correction because spurious features in preference data are correlated with quality signals and with each other, making single-axis interventions prone to redirect rather than remove the bias (Lamparth et al., 2026a). This distinction motivates our approach: mechanistic reward shaping (implemented via linear probe null-space projection) for simple biases (Section 4), and characterization of persistent issues for complex artifacts (Section 5). Beyond the empirical success of our intervention, we provide independent representational evidence for this taxonomy using Iterative Nullspace Projection (Ravfogel et al., 2020) in Section C.9.

## 3.2. Obtaining Linear Activation Probes

We construct linear activation probes that identify bias-encoding directions in the RM's representation space, then project these directions out through null-space projection to obtain debiased reward estimates.

**DiffMean Construction:** To construct our linear activation probes, we use the difference-of-means (DiffMean) method (Marks & Tegmark, 2024), which is the strongest method for concept detection identified by the AxBench benchmark (Wu et al., 2025). For each input, we extract the final-layer hidden state corresponding to the last non-padding token from the base transformer (prior to the reward head). Let $\{\mathbf{h}_i^+\}_{i=1}^{n_+}$ and $\{\mathbf{h}_j^-\}_{j=1}^{n_-}$ denote embeddings of positive and negative examples, respectively. The probe is defined as

$$\mathbf{p} = \text{normalize}\left(\frac{1}{n_+}\sum_i \mathbf{h}_i^+ - \frac{1}{n_-}\sum_j \mathbf{h}_j^-\right).$$

**Null-Space Projection:** After obtaining one or more probes, let $\{\mathbf{p}_k\}_{k=1}^K$ denote the resulting orthonormal probe directions. Given a hidden activation $\mathbf{h}$, we remove all components lying in the span of these probes by projecting onto the orthogonal complement:

$$\mathbf{h}_{\text{null}} = \mathbf{h} - \sum_{k=1}^K \alpha(\mathbf{p}_k^\top \mathbf{h})\,\mathbf{p}_k.$$

Where $\alpha$ refers to the strength of the projection[2]. We refer to this operation as *nulling* a probe from the activation space of an RM, meaning the removal of activation components aligned with the specified probe directions. In many cases, we wish to null multiple probes simultaneously (e.g., to mitigate multiple biases simultaneously) In these cases, we orthogonalize the probes through Gram–Schmidt before applying the intervention.

## 3.3. Datasets and Models

For all bias types, we evaluate five RMs. From the Skywork family, we use Skywork-Reward-V2-Llama-3.1-8B (hereafter Skywork-Llama-8B), Skywork-Reward-V2-Qwen3-8B (hereafter Skywork-Qwen-8B), and Skywork-Reward-V2-Qwen3-0.6B (hereafter Skywork-Qwen-0.6B) (Liu et al., 2025a). Skywork-Reward-V2-Llama-3.1-8B currently leads the RewardBench2 leaderboard, and other tested models (with the exception of the DeBERTa-based RM) are also near SoTA (Malik et al., 2026). Studying multiple RMs that share the Skywork-Reward-V2 training pipeline but differ in base model lets us observe how RM biases vary with the underlying LM. We also evaluate Allen-Llama-3.1-8B-Instruct-RM-RB2 (Malik et al., 2026) (hereafter AllenAI-Llama-8B) and reward-model-deberta-v3-large-v2 (OpenAssistant, 2023) (hereafter DeBERTa) for architectural and finetuning diversity.

Unless otherwise specified, we focus on four reasoning and knowledge benchmarks to identify and evaluate biases: PlausibleQA (Mozafari et al., 2025), BigBench (Srivastava et al., 2023), GSM8K-MC (Zhang et al., 2024; Cobbe et al., 2021), and MMLU (Hendrycks et al., 2021). These datasets span commonsense reasoning, mathematical problem solving, and broad-domain factual knowledge. These datasets are chosen because contemporary RMs achieve non-trivial performance on these tasks without saturating them[3]. This allows us to precisely study how the biases observed relate to task accuracy. For lower complexity biases, we then mitigate the biases and examine the downstream effect of probe-nulling on overall RM performance while showing out-of-distribution generalization using RewardBench2 (Malik et al., 2026; Lambert et al., 2025). Specific dataset processing and splitting details are provided in Table 5.

## 4. Probe Intervention for Simple Biases

### 4.1. Length Bias

Length bias has been documented and addressed in RMs for preference learning (Bu et al., 2025; Chen et al., 2024; Huang et al., 2025; Singhal et al., 2024) as well as direct preference optimization (Park et al., 2024b). However, it

---

[2] $\alpha = 1$ in all of our experiments except confidence calibration.
[3] E.g., these criteria rule out common benchmarks like GPQA.

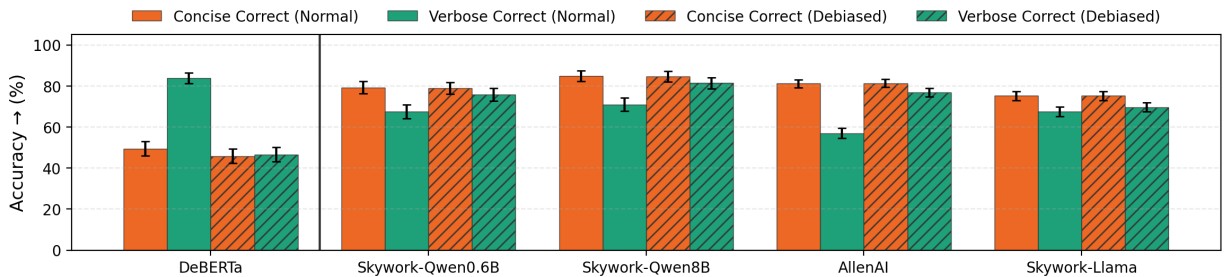

*Figure 1.* RM accuracy selecting correct over incorrect answers on GSM8K with bootstrapped 95% CI. An unbiased RM should show no gap between concise and verbose performance. DeBERTa rewards verbosity (green bar higher) while state-of-the-art models can overcorrect and penalize it (green bar lower). Mechanistic correction via probe nulling closes these gaps without degrading baseline accuracy.

has been noted that this bias is not well-explained by a mere spurious correlation, and is best understood through the decomposition of response desirability and information density (Hu et al., 2025). Moreover, Bu et al. (2025) observe that existing techniques to disentangle length bias often 'overcorrect,' resulting on poor downstream performance.

**Definition:** Here, we measure accuracy on grade school math questions, and vary information density of responses with synthetic data generation. Math questions have the useful property that their answers can be compressed to a minimal length or extended in length while maintaining correctness. This setup can reveal an undesirable preference for information density above correctness (i.e., *length bias*).

**Setup:** We construct a contrastive dataset based on synthetic answers on the GSM8K dataset (Cobbe et al., 2021). For each question, we generate a response triplet: a concise correct answer, an incorrect answer, and a verbose correct answer. To generate incorrect and concise correct answers, we prompt `Llama-3.1-8B-Instruct` to solve each GSM8K problem over 100 trials. We verify the correctness of the answer, and sample an incorrect and a correct response (average word-counts 183.9 and 171.1 words, respectively). Lastly, we prompt the same model to make the correct answer more verbose to obtain the verbose correct answer (average length 477.3 words, verbosity prompt in Section B.2). Our normative target is not global length neutrality, but the absence of length-driven reward differences when answer correctness is held fixed, such that verbosity alone does not cause an RM to prefer an incorrect response. We construct our diff-mean probe as follows:

> **Probe: Length / Verbosity**
>
> **Direction description**: a direction that separates *long/verbose* correct responses from *concise* correct responses, holding the question fixed.

```
User:  {question}
Assistant:
(+) {correct_verbose}
(−) {correct}
```

**Results:** We show the results in Figure 1. We expect an unbiased RM to select the correct response above the incorrect response at the same rate irrespective of the length of the correct response. Thus, a debiasing method should maintain accuracy for concise correct answers, while closing the accuracy gap of verbose correct answers to concise correct answers[4]. We find statistically significant length-related biases across all evaluated RMs, though the direction and severity of the bias varies substantially by architecture. In particular, the DeBERTa-based RM exhibits a classic length bias, preferring verbose correct responses to incorrect responses at a high rate, despite having at-chance performance on the tasks when correct and incorrect answers are unmodified lengths. Meanwhile, several newer state-of-the-art RMs display the opposite behavior, effectively penalizing verbosity. These models are trained on curated datasets to eliminate preference for more verbose responses (Liu et al., 2025a; Malik et al., 2026). However, we find that these efforts can introduce a length bias in the opposite direction, whereby concise but incorrect answers are preferred over correct ones. We demonstrate that our simple mechanistic reward shaping significantly decreases this preference in a controlled setting without degrading accuracy. Further, we find that our method also significantly decreases traditional length bias (preference for longer responses) in the older DeBERTa model that exhibits this artifact. We verify that the debiased RMs do not degrade in performance and that the debiasing works out-of-distribution in Section 4.4.

---

[4]The gap can be positive if the RM rewards verbosity or negative if the RM picks conciseness over correctness

## 4.2. Uncertainty Bias

### 4.2.1. EXPRESSIONS OF UNCERTAINTY

**Definition:** Preference-based RMs are commonly biased against expressing uncertainty (Leng et al., 2025). Normatively, an unbiased *RM with capability on a task* should prefer a direct and correct answer ($C$) to one expressed with uncertainty ($C + U$), and both of these to an incorrect answer ($I$) expressed with or without uncertainty. Finally, unbiased RMs should prefer incorrect answers expressed with uncertainty to direct incorrect answers, i.e., $r(C) \geq r(C + U) \geq r(I + U) \geq r(I)$ for reward $r$.

**Setup:** Here, we prefix answer choices with an expression of uncertainty to elicit directness bias in RMs. Then, we evaluate adherence to the normative ranking spelled out above. Lastly, we split by model capability on the task to evaluate whether the RMs preference for uncertainty is related to its true uncertainty. We construct diff-mean probes with and without this expressed uncertainty as follows:

> **Probe: Uncertainty**
>
> **Direction description**: a direction that captures *hedging* ("I'm not entirely sure...") versus direct declarative answering, with identical question and answer content.
>
> ```
> User:  Question:  {question}
> Assistant:
> ```
> **(+)** I'm not entirely sure, but I think the answer is {answer}.
> **(−)** The answer is {answer}.

**Results:** We measure the incidence of preference for or against uncertainty in a variety of contexts, with and without our debiasing approach. First, we measure how often a model prefers an incorrect answer expressed with uncertainty to an incorrect answer without uncertainty. We find that all models significantly prefer direct but incorrect answers. Even in relation to correct answers, we find that models accuracy drops by an average of 22.6% when the correct answer contains uncertainty (see by-model uncertainty penalties in Table 8 and Figure 2).

We find that our method significantly reduces this bias in all models tested, increasing preference for uncertainty when answers are incorrect, and increasing accuracy when correct answers are expressed with uncertainty. Moreover, while our method increases preference for uncertainty above directness broadly, we find that the strength of this preference is modulated by the model's true uncertainty. For every model, when the RM correctly ranks the answers without any uncertainty injected, it prefers direct answers to uncertain ones after the intervention. And when the model is incorrect, it prefers uncertain answers to direct answers. All of these results can be seen visualized in Figure 2 and

per-dataset in Table 7. We also verify out-of-distribution debiasing performance in Section 4.4.

### 4.2.2. CALIBRATION VIA REWARD SHAPING

**Definition:** Beyond expression of uncertainty, the verbalized confidence by language models is often not reflective of actual model uncertainty (Sun et al., 2025). Leng et al. (2025) show this by recording RM calibration with different confidence ratings appended to possible responses. Here, we adopt a very similar prompt to observe the effectiveness of bias removal at enhancing calibration.[5] An ideal RM should prefer higher confidence declarations if and only if the underlying answer is correct, and prefer lower confidence when the answer is incorrect.

**Setup:** We conduct two evaluations for each example across our datasets to measure calibration. In the first, we determine the frequency with which the model selects the correct response (by assigning it the highest reward). Then, we determine, for each correct response, what confidence suffix the RM prefers for that question-answer pair out of 'confidence: $x \, \forall x \in \{$low, medium, high$\}$'. Lastly, we calculate the Spearman correlation between model rating and model correctness. A calibrated model should have a higher correlation, with confidence being more predictive of model correctness, and thus more trustworthy.

Then, we construct probes associated with selecting each confidence level, and null out each direction sequentially after Gram-Schmidt orthogonalization. When an RM is strongly biased, we expect this probe to capture spurious preference for high confidence declarations, and debiasing to remove this preference without affecting underlying calibration, effectively improving correlation. When RMs are not strongly biased in this way, we expect confidence declarations to be rightly entangled with actual model confidence, and the probe to potentially remove some of this information. Since this probe may or may not remove useful information, we also ablate over the strength of the subtraction of the projection of our activation onto the probe from the activation. Here, $\alpha$ represents this factor (e.g. $\alpha = 1$ represents the standard null-space projection demonstrated in Section 3.2).

**Results:** All Spearman correlations are available in Table 1. We find that the most calibrated baseline model has a Spearman correlation of 0.35. For both Skywork-Qwen models as well as the AllenAI Llama-based model, probing significantly increases calibration. The mechanistic intervention doubles the calibration of Skywork-Qwen-8B, and the resultant model is the most calibrated model tested. DeBERTa does not exhibit high correlation in any setting, likely due to high uncertainty throughout the evaluation set. Interestingly,

---

[5]Leng et al. (2025) use a confidence scale with range 0-10. Here, we opt. for 'low', 'medium', and 'high' to avoid biases we find some models have pertaining to specific Likert selections.

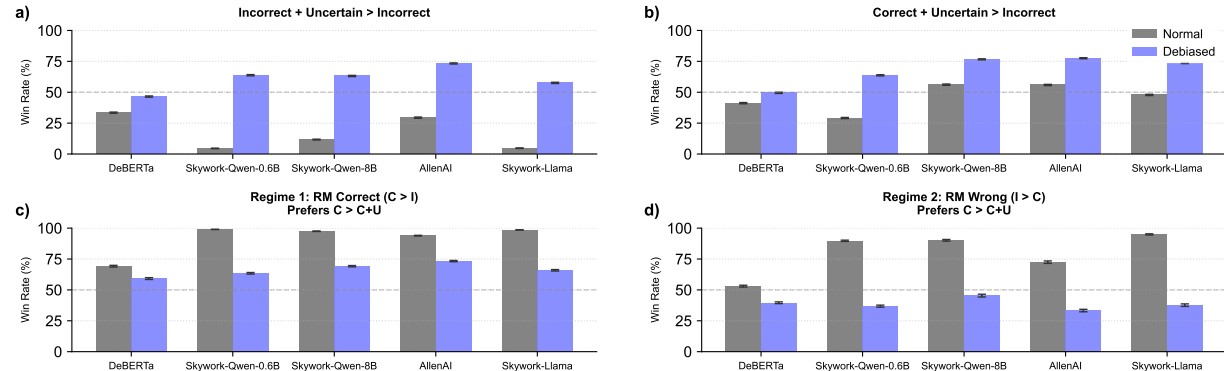

*Figure 2.* Win rates comparing responses with and without verbalized uncertainty, before and after debiasing with bootstrapped 95% CI. (a), (c), and (d) show that, before debiasing, models prefer direct answers irrespective of correctness. (b) shows that this frequently causes RMs to prefer incorrect answers to correct answers expressed with uncertainty. In all experiments, debiasing reduces over-penalization of uncertainty. (c) and (d) show increasing preference for uncertainty when answers are incorrect while preserving preference for direct answers when the model is correct.

| Model | Baseline | $\alpha = 0.5$ | $\alpha = 1.0$ | $\alpha = 1.5$ |
|---|---|---|---|---|
| Skywork-Qwen-8B | $0.182_{\pm 0.007}$ | $0.328_{\pm 0.008}$ | $\mathbf{0.386_{\pm 0.008}}$ | $0.008_{\pm 0.006}$ |
| Skywork-Qwen-0.6B | $0.068_{\pm 0.008}$ | $0.069_{\pm 0.008}$ | $\mathbf{0.146_{\pm 0.008}}$ | $-0.000_{\pm 0.009}$ |
| AllenAI-Llama-8B | $0.271_{\pm 0.007}$ | $\mathbf{0.332_{\pm 0.008}}$ | $0.102_{\pm 0.009}$ | $0.055_{\pm 0.005}$ |
| Skywork-Llama-8B | $0.350_{\pm 0.009}$ | $\mathbf{0.353_{\pm 0.009}}$ | $0.129_{\pm 0.008}$ | $-0.005_{\pm 0.009}$ |
| DeBERTa | $0.001_{\pm 0.008}$ | $0.008_{\pm 0.008}$ | $0.035_{\pm 0.008}$ | $\mathbf{0.037_{\pm 0.009}}$ |

*Table 1.* Spearman correlation between verbalized model confidence (low, medium, high) and answer correctness across all datasets for each model with 95% confidence intervals bootstrapped across all evaluated examples. A larger correlation indicates a better calibration. Varying the probe strength $\alpha$, we can achieve significant increases of the correlation for each model above the unmodified RMs (baseline).

both Llama8B-based models have relatively high correlation before the intervention. In these cases, we find that a weaker null-projection factor is more suited to maintaining (and even increasing) correlation between verbalized confidence and accuracy.

### 4.3. Position Bias

**Definition:** Position bias is a well-studied phenomenon whereby language models are biased toward selecting answers based on their position in a list of options (Shi et al., 2025; Bito et al., 2025), but so far unstudied in RMs. Although the bias has been widely studied in the generative language models, we demonstrate here that position bias is also observable in strong RMs.

**Summary:** Our full setup and results are in Section C.3. In summary, we evaluate position bias in both multiple-choice question answering (MCQA) and free-form settings. In the MCQA format, we systematically vary the position of correct answers across positions A to D. In the free-form setting, we present answer options as comma-separated continued text and measure preference based on whether the correct answer appears first or last. We observe statistically significant position bias across all evaluated RMs, with deviations ranging from 2 to 28% across answer positions. DeBERTa exhibits strong bias toward the last answer choice, while

SOTA models consistently prefer at least one answer option over others. Mechanistic reward shaping via probe nulling significantly reduces positional variance for three models in the MCQA and free-form setting. Detailed results, including per-dataset breakdowns and the effectiveness of debiasing across both settings, are also provided in Appendix C.3.

### 4.4. Out-of-Distribution Performance After Debiasing

To ensure that our mechanistic interventions do not degrade RM performance, we evaluate our mechanistic interventions along three axes. First, we verify that interventions preserve *ranking performance*, i.e., the ability to correctly rank chosen over rejected completions, as measured by RewardBench2 and prior benchmarks. Second, we examine *reward calibration and out-of-distribution generalization*, asking whether probes trained on narrow data reshape the reward landscape as intended on broader held-out distributions rather than introducing dataset-specific artifacts. Third, and closest to deployed RLHF use, we test whether the length probe transfers to best-of-N (BoN) candidate selection on held-out AlpacaEval and GSM8K prompts, the setting that ultimately drives policy outputs.

Table 2 shows RewardBench2 accuracies averaged across models before and after each intervention. All interventions demonstrate statistical non-inferiority to baseline at a

$\delta = 5$ percentage point margin, chosen to match the empirical spread among top-ranked RMs on the RewardBench2 leaderboard.[6] This confirms that bias mitigation does not significantly degrade the RM ability to distinguish chosen from rejected completions across diverse tasks spanning safety, reasoning, and chat domains. Detailed per-model results are shown in Table 11.

While RewardBench2 validates ranking preservation, it does not measure whether interventions affect reward magnitudes or distributions. To examine this, we analyze changes in mean chosen rewards, mean rejected rewards, and their difference (Table 12). We observe consistent patterns suggestive of successful debiasing: interventions generally narrow the mean reward difference between chosen and rejected samples across models with statistical significance, while also decreasing the mean chosen reward for three models. This narrowing occurs primarily through shifts that bring outlier rewards toward the distribution center. This pattern is consistent with removing spurious correlations that artificially inflate or deflate rewards based on surface features rather than true quality.

A critical question is whether probes trained on narrow domains[7] generalize to the diverse prompt-completion pairs in RewardBench2. To examine this, we plot reward distributions against completion lengths for all models before and after the length correction probe in Figures 4 to 8. We see that the probe debiases the DeBERTa-based RM, which exhibits classic length bias without specific training-time corrections, as we observe significant decorrelation (Spearman $r_s$ decreases from 0.611 to 0.067, 95% CI non-overlapping) with the probe appropriately reducing rewards for longer sequences. For newer models trained to avoid length bias but exhibiting overcorrection (preferring concise incorrect answers), the probe adds slight positive correlation (e.g., Skywork-Llama-8B $r_s$ increases from 0.267 to 0.305), primarily affecting longer responses. These effects match our theoretical predictions for each model class and demonstrate that the probe captures generalizable, transferable features of length bias rather than dataset-specific artifacts.

We also evaluate the effect of the uncertainty probe on OOD RewardBench2 using an LLM-as-a-judge approach. To do this, we prompt `GPT-5.1` to rate (1-10) how much each response expressed uncertainty among the responses where the unmodified and uncertainty-debiased RMs disagree. The probe-preferred response is rated as expressing significantly higher uncertainty ($p < 0.001$), confirming that the probe works as expected OOD (further details in Section C.5).

---

[6]The leaderboard currently shows only a $\sim$5pp spread among the top-5 SOTA models (top-10 span $\sim$8pp).

[7]Especially for the length debiasing, which only used one dataset (GSM8K), while the uncertainty and position bias probes were constructed and verified across multiple datasets and tasks.

| Condition | Baseline | Length | Position | Uncertainty | Combined |
|---|---|---|---|---|---|
| Mean Acc. (%) | 70.1 | 69.3* | 69.3* | 70.1* | 69.3* |

*Table 2.* RewardBench2 averaged accuracies for each bias type (*) indicates non-inferiority to baseline under a one-sided paired non-inferiority t-test with margin $\delta = 5$ percentage points. All non-inferiority p-values are $< 0.001$.

Finally, we evaluate the length probe under best-of-N selection on held-out AlpacaEval and GSM8K prompts in Tables 19 and 20. With probes fit on a disjoint 293-prompt pool and evaluated on a 512-prompt test pool with 64 candidates per prompt, mechanistic reward shaping reduces the mean absolute within-prompt reward–length correlation across five RMs from 0.10 at baseline to 0.04 and improves AlpacaEval length-controlled win rate on four of five RMs, outperforming the global calibration of Huang et al. (2025) on both diagnostics. On held-out GSM8K with 64 generations per question, mechanistic reward shaping moves the within-prompt correlation toward zero on all five RMs (mean absolute 0.076 to 0.007) and improves mean best-of-N accuracy from 62.1% to 62.8%. The probe therefore transfers from the construction set to the prompt-conditioned candidate-ranking setting that ultimately drives policy outputs.

## 5. Unsolved Reward Proxies

We turn to **high-complexity biases**: RM failure modes where RM signal is co-linear with the bias in activation space, and therefore observed biases are resistant to simple linear interventions.

### 5.1. User Agreement (Sycophancy) Bias

Sycophancy is a complex phenomenon that has been defined as the tendency of language models to excessively agree with or flatter users, often at the expense of factual accuracy or ethical considerations (Malmqvist, 2025). While sycophancy has been documented in RMs and RLHF-trained language models (e.g., Perez et al., 2023), we evaluate whether current state-of-the-art RMs still exhibit this behavior and to what extent.

**Definition:** Here, we study a simplified model of *sycophancy based on user agreement*. It has been shown that preference models often prefer assistant responses that agree with users above correct responses (Sharma et al., 2024). We adopt the terminology of Fanous et al. (2025) who distinguish progressive sycophancy, where the user suggestion leads to correctness, and regressive sycophancy, where the user suggestion leads to incorrect answers. We use a similar methodology to these prior works by adding user-suggested correct and incorrect answers to the user message, and observing how frequently the preference model prefers the correct answer ("agreement rate").

| Reward Model | Sycophancy Type | |
| --- | --- | --- |
| | Regressive ($\downarrow$) | Progressive ($\uparrow$) |
| DeBERTa | $0.633 \pm 0.009$ | $0.602 \pm 0.009$ |
| Skywork-Qwen-0.6B | $0.431 \pm 0.008$ | $0.840 \pm 0.006$ |
| Skywork-Qwen-8B | $\mathbf{0.237 \pm 0.005}$ | $0.959 \pm 0.003$ |
| AllenAI-Llama-8B | $0.583 \pm 0.007$ | $\mathbf{0.996 \pm 0.001}$ |
| Skywork-Llama-8B | $0.308 \pm 0.008$ | $0.893 \pm 0.006$ |

*Table 3.* Baseline rates of regressive and progressive sycophancy (i.e., agreement with an incorrect or correct user response, respectively) across models with 95% bootstrapped uncertainty intervals. Filtered for questions the respective RM can correctly answer.

Unlike prior work evaluating sycophancy in RMs (Sharma et al., 2024; Hong et al., 2025), we filter our evaluation set to include only prompts where the RM correctly identifies the right answer in the absence of user suggestions by giving the highest reward to the correct option among distractors. This filtering allows us to isolate cases where agreeing with an incorrect user suggestion is objectively problematic, rather than potentially defensible uncertainty. We exclude cases where the RM initially answers incorrectly, as user guidance in such scenarios could be considered reasonable model behavior.

**Results:** The results are shown in Table 3. All five tested RMs show statistically significant sycophancy across datasets, though the four newer state-of-the-art models substantially outperform the older DeBERTa-based RM. In the desirable case, where the RM "knows" the answer if directly evaluated and the user identifies the correct answer, agreement rates should reach 100%, yet the best-performing models (Skywork-Qwen-8B and AllenAI-Llama-8B) fall short of this target. More problematically, when users suggest incorrect answers and agreement rates should be 0%, all models show agreement rates exceeding 24%, with DeBERTa and AllenAI-Llama-8B reaching 63% and 58% respectively.

We also evaluate a linear intervention for user agreement (see Figure 9 for detailed results). We find that the intervention cannot decrease faulty-agreement without decreasing helpful-agreement or other trade-offs, suggesting this bias is complex in that user agreement is co-linear with useful signals in RM activation space.

### 5.2. Model-Style Reward Correlation Bias

RMs could inadvertently latch onto the stylistic signatures of the models that generated (synthetic) preference data or were annotated in their training data, so they may reward "familiar" model styles. Prior work (e.g., Malik et al., 2026) also observed that RLHF performance improves when RMs are paired with the same base language model to be trained during RLHF, which could also be partially attributed to model-style sensitivity.

**Definition:** We define *model-style reward sensitivity* as a po-

tential correlation between a scalar reward $r$ and how much a completion's likelihood under a given generative language model $P_m$ deviates from a cross-model baseline. Thus, non-zero correlation values would indicate that rewards systematically increase or decrease for completions that are relatively more "in-distribution" for model $m$ compared to other models. To this end, we calculate the (negative) per-byte cross-entropy for a generative language model $P_m$ for a given prompt-completion pair $(x, y)$ over tokens $t$ as

$$s_m(y \mid x) = -\text{NLL}_m(y \mid x) \cdot [\text{bytes}(y)]^{-1} \quad (1)$$

with

$$\text{NLL}_m(y \mid x) = -\sum_{t=1}^{T} \log P_m(y_t \mid x, y_{<t}).$$

We normalize with $\text{bytes}(y)$ to avoid inducing effects from combining values form models using different tokenizers later. Similar to perplexity, cross-entropy in Equation (1) also measures how "common" a completion is given the prompt and does not capture model-specific style alone. Thus, we construct a *panel-relative measure* for cross-entropy as

$$\Delta s_m(x, y) = s_m(y \mid x) - s_{\text{panel}}(y \mid x) \quad (2)$$

with the panel aggregated (negative) per-byte cross-entropy as $s_{\text{panel}}(y \mid x) = \text{agg}_{p \sim \mathcal{P} \setminus \{m\}} s_p(x, y)$ aggregated over a representative list of (other) generative language models. As an aggregation function, we use the sample median $\tilde{s}_i$ as default for more robustness to potential outliers. Finally, we calculate the Spearman correlation $\rho_s$ between the reward from RM $i$ and the panel-relative cross entropy as $\rho_s(r_i(x, y), \Delta s_m(x, y))$. Statistically significant non-zero correlations would indicate a reward contamination by either rewarding or punishing specific model-styles.

**Setup:** We use ten generative language models to calculate cross-entropy from the Gemma family (*Gemma-2-2b-it*, *Gemma-2-9b-it*, *Gemma-3-12b-it*) (Gemma Team, 2024; 2025), Llama family (*Llama-2-7b-chat-hf*, *Llama-2-13b-chat-hf*, *Llama-3.1-8B-Instruct*) (Touvron et al., 2023; Grattafiori et al., 2024), and the Qwen family (*Qwen2.5-0.5B-Instruct*, *Qwen2.5-7B-Instruct*, *Qwen3-0.6B*, *Qwen3-8B*) (Yang et al., 2024; Qwen Team, 2025). Besides including the base models of some of the tested RMs, our model selection is guided by broad model representation. We use the *tulu-3-wildchat-reused-on-policy-8b* dataset, which comprises prompts from *WildChat* (Zhao et al., 2024a) and generations of 24 language models based on, providing a rich distribution of model completions grounded in realistic user queries and text lengths. Each prompt comes with a chosen and a rejected completion. In our analysis, we use $n = 2400$ prompt-chosen-rejected tuples, leading to 4800

| REWARD MODEL $i$ | MEAN($|\rho_s(r_i, \Delta s_m)|$) |
|---|---|
| ALLENAI-LLAMA-8B | 0.08 |
| SKYWORK-LLAMA-8B | 0.11 |
| SKYWORK-QWEN-8B | 0.09 |
| SKYWORK-QWEN-0.6B | 0.12 |
| DEBERTA | 0.10 |

*Table 4.* Mean absolute Spearman correlations between panel-relative (negative) per-byte cross-entropy $\Delta s_m(x, y)$ and rewards $r_i(x, y)$ across the tested ten language models. See Section C for detailed results and statistical uncertainties.

evaluated $(r_i(x, y), \Delta s_m(x, y))$ data points for each $(i, m)$ pair.

**Results:** An overview of the results is shown with the average absolute correlation coefficients across models $m$ for each RM $i$ in Table 4 and the detailed results with statistical uncertainties are shown in Tables 14 to 18 in Section C. For all tested RMs, we observe weak but statistically significant correlations between panel-relative cross-entropy and rewards, demonstrating that RM rewards are contaminated by rewarding or punishing specific model-styles. While the average absolute correlations are only around 0.1 in Table 4, each tested RM shows correlations for individual language models of $\pm 0.2$ to 0.4. Using $\rho_s^2$ as a heuristic measure of variance explained in rank space, these effect sizes correspond to *approximately* 1% on average and 4-16% for individual models of the variance in reward ranks being associated with model-style sensitivity. This sensitivity affects top performing SOTA RMs similarly to the less competitive and older DeBERTa-based RM, directly impacting widely used open-source preference datasets. Implicated model families (Llama, Qwen, Gemma) are heavily represented in, e.g., ChatbotArena (Chiang et al., 2024), LMSYS-Chat-1M (Zheng et al., 2024), and AllenAI's Tulu 2.5 Preference Data (Ivison et al., 2024), suggesting that RMs trained on such data may reward recognizable stylistic signatures rather than genuine quality. Consequently, RM-policy pairing decisions and training data curation should account for potential style-driven reward inflation. We discuss further implications in Section 6 and the impact statement.

**Ablations:** We find statistically significant correlations, even if we use Pearson instead of Spearman correlation, aggregate by taking the mean instead of the median in Equation (2), use different normalizations for $s_m$ in Equation (1) (number of tokens and characters instead of bytes), using a variety of subsets of the models comprising the panel (e.g., only using RM base or newer models or removing seemingly strongly positive reward-correlated models like *Qwen2.5-7B-Instruct*), and using models in "thinking mode" or not through API keywords.

## 6. Discussion and Limitations

In this work, we show that simple biases still largely persist in state-of-the-art RMs, and can be mitigated by simple linear interventions. Further, we show that there remain complex unsolved biases that are entangled with RM judgments. We construct our length correlation probes and mitigation on GSM8K, but evaluate OOD on RewardBench2 and on AlpacaEval prompts with BoN selection. However, length-vs-correctness tradeoffs in non-verifiable open-ended generation remain harder to measure directly. The magnitude and the sign of each bias vary across RMs (Figure 1, Table 1), and the quality of the linear fix depends on both the target RM and the probe construction dataset. Smaller or differently trained RMs can encode the same bias along multiple directions (e.g., see Table 21 and Table 1). Probe construction dataset also matters in open-ended settings, motivating our use of fresh AlpacaEval-derived probes for AlpacaEval best-of-N (Table 19) rather than reusing the GSM8K-constructed probe used elsewhere.

Across most biases, our study is limited to verifiable tasks, despite the reality that many open-ended tasks are difficult to verify yet crucial. We also do not examine the extent to which biases are learned by base models during the process of RLHF. Though RM evaluations are helpful, other parameters and aspects of post-training (base-policy, algorithm choice, sampling, etc.) may also play a significant role in bias proliferation.

## Acknowledgements

We are grateful to Julie Kallini and Houjun Liu for giving feedback on an earlier version of this work, and to Gabriela Aranguiz Dias for participating in helpful discussions. Max Lamparth is supported through a grant from Coefficient Giving (formerly Open Philanthropy), Stanford's Hoover Institution Tech Policy Accelerator, and the Stanford Intelligent Systems Laboratory.

## Impact Statement

**General Implications:** Our work contributes to a growing body of evidence that even state-of-the-art reward models exhibit biases that may affect the efficacy and transparency of language model alignment. By empirically characterizing these biases and successfully mitigating some of them, this work may help practitioners build more robust alignment pipelines. Further, we contribute to evidence that many biases are complex downstream consequences of decisions made by developers, some of which may not be visible without careful measurement. This carries implications for the design of governance of systems that are released to the public.

**Implications of sycophancy and overconfidence results:** Sycophancy and overconfidence pose significant risks for downstream applications, with potential consequences for user trust, safety, and decision-making (Wen et al., 2025; Grabb et al., 2024; Cheng et al., 2026b;a). Addressing this issue is essential yet non-trivial, as naive interventions risk overcorrecting and suppressing desirable agreement behaviors. Our study of user agreement sycophancy does not encompass the full range of forms or consequences of sycophancy.

**Implications of model style sensitivity:** Model-style reward sensitivity is problematic because it can push RLHF to optimize for "familiar" phrasing rather than genuine quality, systematically advantaging some model dialects and confounding both transfer and evaluation. Because these model families dominate widely used open-source preference corpora (Section 5.2), this bias is likely to affect any RM trained on such data. It is also plausible, that reward models are also sensitive to model writing styles from the OpenAI and Anthropic families, given that they are represented in common preference corpora (though we cannot verify this without logit access).

Practitioners should therefore consider style-driven reward inflation when selecting RM-policy pairings and curating preference data, since a policy trained with an RM biased toward its own base-model style may receive inflated reward signals irrespective of output quality. However, addressing this bias is difficult because the signal is entangled with content and can arise from multiple, prompt-dependent, non-linear sources, so a simple global debiasing (e.g., a linear probe) cannot work. This effect is probably even more amplified when doing RLHF in more complex domains where we have to rely on human expert feedback, e.g., mental health Chatbots for therapy sessions or clinical decision-making (Grabb et al., 2024; Gabriel et al., 2024; Moore et al., 2025; Lamparth et al., 2026b).

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

# A. Large Language Model Usage Outside of Experiments

We used large language models sparingly in the creation process of this work. In particular, we used it for some minor writing polish and feedback (e.g., "Is this section written clearly or are there overly wordy sections?") or to provide minor writing aid (e.g., Latex table formatting).

We take full responsibility for all content in this paper, including any content generated by large language models that might be construed as plagiarism or scientific misconduct. We did not use large language models to generate substantive text or research content for this publication

# B. Additional Experiment Details

## B.1. Dataset Information

*Table 5.* Summary of evaluation datasets, their general modifications for auditing, and split strategies.

| Dataset | Authors | Dataset Processing and Splitting | Probe Size | Test Size |
|---|---|---|---|---|
| **PlausibleQA** | Mozafari et al. (2025) | QA dataset with choices sorted by plausibility. We select the 3 most plausible incorrect answers and the target answer to construct multiple-choice questions. | 500 | 738 |
| **BigBench** | Srivastava et al. (2023) | We use the 18 multiple-choice tasks from BigBench-lite. Since the dataset is not balanced, we sample a maximum of 500 examples from each into our evaluation set (and 100 from each into our probe). We use a shared probe across all sub-tasks. | 1172 | 5567 |
| **GSM8K-MC** | Zhang et al. (2024) | A multiple-choice adaptation of GSM8K (Cobbe et al., 2021). We use the train split for probe construction and the test split for evaluation, with prompts modified via formatted templates. | 500 | 7,468 |
| **MMLU** | Hendrycks et al. (2021) | Covers 57 graduate-level subjects in MCQ format. We use the train split and use 500 samples to segregate a probe dataset. | 500 | 13,542 |
| **RewardBench** | Lambert et al. (2025) | Benchmark comprising safety, reasoning, and chat categories. We use this dataset solely for studying adverse effects of probe-nulling. | N/A | 2,985 |
| **Tulu-3 WildChat** | AllenAI (2024) | Prompts from the WildChat dataset (Zhao et al., 2024a) with completions (chosen, rejected) from 24 different generative language models. We use this dataset solely to study correlations between reward and language model-specific styles. | N/A | 4800 |

### B.2. Verbosity Prompt

> **Verbose Rewrite Prompt**
>
> *Rewrite the following solution in a much more verbose way. Add more explanation, detail each step thoroughly, and include additional context. Make it at least 2–3 times longer while keeping the same answer.*
>
> **Original solution:**
>
> {concise_solution}
>
> **Verbose rewrite:**

# C. Additional Results

## C.1. Length Bias - Additional Results

*Table 6.* Length bias: Accuracy rate (%) when choosing correct answer over incorrect one. Format: baseline / nulled. Higher is better. Bold indicates significant difference ($p < 0.05$).

| Model | GSM8K |
| --- | --- |
| *Concise Correct Accuracy (higher is better)* | |
| DeBERTa | 49.5 / 45.8 |
| Skywork-Qwen-0.6B | 79.3 / 78.9 |
| Skywork-Qwen-8B | 84.9 / 84.5 |
| AllenAI-Llama-8B | 81.2 / 81.9 |
| Skywork-Llama-8B | 75.2 / 75.3 |
| *Verbose Correct Accuracy (higher is better)* | |
| DeBERTa | **83.7** / 46.6 |
| Skywork-Qwen-0.6B | 67.5 / **75.9** |
| Skywork-Qwen-8B | 71.0 / **81.4** |
| AllenAI-Llama-8B | 56.2 / **75.5** |
| Skywork-Llama-8B | 67.7 / 69.8 |

## C.2. Uncertainty and Overconfidence - Additional Results

*Table 7.* Uncertainty bias metrics (%). Format: baseline / nulled. Bold indicates significant difference ($p < 0.05$).

| Model | GSM8K-MC | MMLU | PlausibleQA | BigBench |
|---|---|---|---|---|
| *I+U > I (higher is better)* | | | | |
| DeBERTa | 26.7 / **46.0** | 37.6 / **48.9** | 31.2 / **43.6** | 38.9 / **46.7** |
| Skywork-Qwen-0.6B | 1.8 / **52.1** | 2.9 / **82.9** | 6.0 / **49.6** | 10.3 / **54.2** |
| Skywork-Qwen-8B | 3.3 / **65.8** | 6.1 / **69.7** | 23.9 / **59.7** | 16.8 / **60.6** |
| AllenAI-Llama-8B | 5.6 / **57.2** | 28.3 / **80.4** | 62.6 / **76.6** | 8.5 / **64.1** |
| Skywork-Llama-8B | 0.7 / **60.2** | 2.4 / **47.1** | 8.3 / **76.0** | 10.0 / **55.7** |
| *C+U > I (higher is better)* | | | | |
| DeBERTa | 32.0 / **50.8** | 46.0 / **50.5** | 40.1 / **46.1** | 42.3 / **48.8** |
| Skywork-Qwen-0.6B | 25.4 / **58.6** | 30.9 / **78.2** | 24.4 / **46.8** | 31.7 / **54.4** |
| Skywork-Qwen-8B | 47.1 / **78.1** | 55.6 / **79.9** | 62.8 / **73.5** | 46.7 / **65.2** |
| AllenAI-Llama-8B | 35.1 / **68.2** | 56.8 / **81.3** | 77.7 / **80.9** | 38.6 / **67.7** |
| Skywork-Llama-8B | 26.5 / **68.1** | 46.3 / **73.0** | 66.8 / **81.1** | 43.0 / **66.5** |
| *RM Correct: C > C+U* | | | | |
| DeBERTa | 78.6 / **61.9** | 64.1 / **58.3** | 71.1 / **59.8** | 65.4 / **56.7** |
| Skywork-Qwen-0.6B | 100.0 / **66.8** | 99.5 / **62.7** | 97.5 / **65.1** | 98.1 / **58.3** |
| Skywork-Qwen-8B | 100.0 / **73.6** | 99.8 / **69.6** | 95.0 / **72.0** | 91.4 / **55.5** |
| AllenAI-Llama-8B | 99.9 / **73.3** | 95.8 / **75.1** | 84.6 / **76.5** | 98.4 / **62.4** |
| Skywork-Llama-8B | 100.0 / **59.9** | 99.7 / **65.4** | 98.8 / **74.3** | 93.1 / **58.6** |
| *RM Wrong: C > C+U* | | | | |
| DeBERTa | 61.4 / **40.1** | 48.6 / **39.2** | 55.0 / **37.7** | 50.9 / **43.6** |
| Skywork-Qwen-0.6B | 99.4 / **44.6** | 95.4 / **41.6** | 82.3 / **27.3** | 85.8 / **40.7** |
| Skywork-Qwen-8B | 99.7 / **60.8** | 98.7 / **49.7** | 78.9 / **32.4** | 88.7 / **49.4** |
| AllenAI-Llama-8B | 99.6 / **47.2** | 82.3 / **40.6** | 23.5 / **13.4** | 90.5 / **32.5** |
| Skywork-Llama-8B | 100.0 / **50.8** | 98.6 / **33.7** | 87.2 / **23.1** | 91.4 / **47.3** |

*Table 8.* Average uncertainty penalty (pp) across datasets: Baseline vs. Nulled.

| Condition | DeBERTa | Skywork-Qwen-0.6B | Skywork-Qwen-8B | AllenAI-Llama-8B | Skywork-Llama-8B |
|---|---|---|---|---|---|
| Baseline Penalty | 9.4 | 30.6 | 23.1 | 22.6 | 27.1 |
| Nulled Penalty | -3.6 | -1.5 | 1.8 | 0.5 | 0.8 |

## C.3. Position Bias - Additional Results

**Definition:** Position bias is a well-studied phenomenon whereby language models are biased toward selecting answers based on their position in a list of options (Shi et al., 2025; Bito et al., 2025), but so far unstudied in RMs. Although the bias has been widely studied in the generative language models, we demonstrate here that position bias is also observable in strong RMs.

**Setup:** We study position bias in a multiple-choice and a free-form setting, as RMs could express position bias when answer choices are provided in a list in text without the scaffolding of a multiple choice construction, e.g., as in popular benchmarks. This setting is potentially more natural for the instruction-tuning setting. Although RMs are rarely trained or evaluated on multiple-choice data, we argue that positional invariance is relevant in instruction-following settings, where systematic positional biases can be taught to more capable policies during RLHF. For the multiple choice format, we construct our probe and dataset as follows:

---

**Probe: Position (Multiple Choice)**

**Direction description**: a basis of directions that capture "the selected answer is at position $P$" for $P \in \{A, B, C, D\}$, orthonormalized via Gram–Schmidt.

---

*Table 9.* Position bias (MCQ): Std. dev. of accuracy (%) across positions A/B/C/D. Format: baseline / nulled. Lower is better. Bold indicates significantly lower ($p < 0.05$).

| Model | GSM8K-MC | MMLU | PlausibleQA | BigBench |
|---|---|---|---|---|
| DeBERTa | 27.7 / **4.3** | 15.0 / **7.7** | 26.7 / **2.7** | 19.8 / **10.9** |
| Skywork-Qwen-0.6B | 17.2 / 16.0 | 1.4 / 1.5 | 14.3 / 13.5 | 6.1 / 6.9 |
| Skywork-Qwen-8B | 7.8 / **6.4** | 2.0 / 2.0 | 9.2 / **6.8** | 2.4 / 2.2 |
| AllenAI-Llama-8B | 10.8 / **7.9** | 5.3 / **4.2** | 10.8 / **8.0** | 6.1 / 7.5 |
| Skywork-Llama-8B | 7.6 / 7.7 | 0.8 / 0.9 | 5.9 / 5.9 | 3.8 / 3.8 |

```
User:
Question:  {question}

A. {choice_A}
B. {choice_B}
C. {choice_C}
D. {choice_D}

Assistant:
(+) The answer is {P}.  {target}.  (target at position P)
(-) The answer is {Q}.  {target}.  (target at position Q ≠ P)
```

For free-form evaluation, we rearrange the choices into a paragraph-form body of text as such:

**Probe: Position (Free-form)**

**Direction description**: a direction that separates "correct answer listed *first*" from "correct answer listed *last*" in a comma-separated list of options.

```
User:
{question}

The answer is either {option_1}, {option_2}, {option_3}, or {option_4}.

Assistant:
(+) {correct} (correct listed first)
(-) {correct} (correct listed last)
```

In both settings, we evaluate each data example with the correct answer inserted each of the evaluated positions and record accuracy.

**Results:** In both the multiple choice and freeform settings, we find that position bias is present across all evaluated RMs and sometimes substantial, even in very strong RMs. Figure 3 demonstrates this bias: each bar shows the accuracy of the model across all evaluated questions when the correct answer choice is exclusively in the labeled position. A model without positional bias would have all bars within an experiment with the same height. Instead, we see that within experiments, models are accurate at different rates with the correct answer occupying certain answer choices. Further, this bias is RM-dependent, and unevenly expressed across domains with greater effect size in grade school math than other evaluated domains (see Table 10 for dataset-specific biases).

Applying mechanistic reward shaping significantly mitigates positional bias in both multiple-choice and freeform contexts, though it does not fully eliminate the bias, nor does it have the same effect for each model. For DeBERTa, we find and address extremely pronounced bias towards the last answer choice. In all other models, we find the opposite – significant preference for the first answer choice above the last answer choice. We decrease this bias in both settings for Skywork-Qwen-0.6B, in multiple-choice for AllenAI-Llama-8B, and in freeform for Skywork-Llama-8B. In the cases where the effect is not significant, we see small trends towards bias mitigation. These results can be seen in Figure 3.

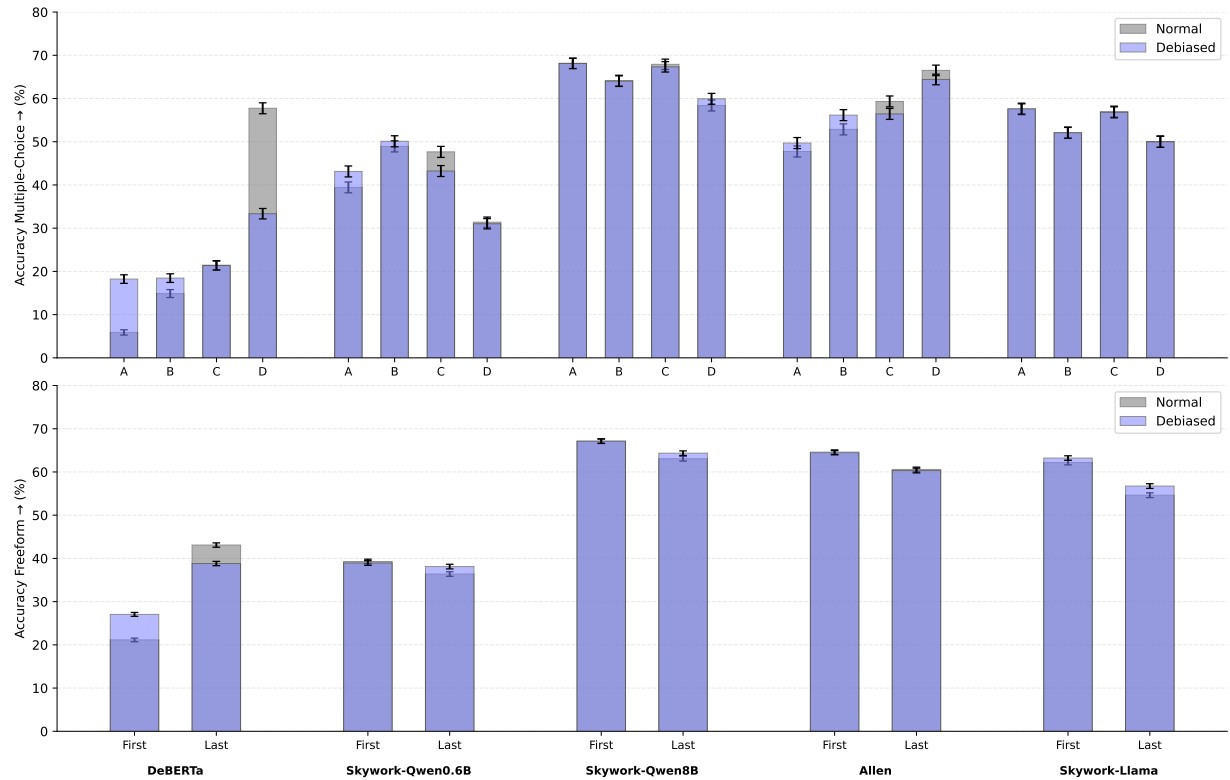

*Figure 3.* Position bias is present in all tested models. Nulling probes representing positional information position mitigates this bias in many cases. Error bars are bootstrapped across all evaluated data.

*Table 10.* Position bias (Freeform): Accuracy gap (%) between first and last position. Format: baseline / nulled. Lower is better. Bold indicates significantly lower ($p < 0.05$).

| Model | GSM8K | MMLU | PlausibleQA | BigBench |
|---|---|---|---|---|
| DeBERTa | 37.2 / **8.7** | 19.1 / **18.2** | 12.8 / **2.3** | 24.1 / **21.3** |
| Skywork-Qwen-0.6B | 4.4 / **2.4** | **3.3** / 5.5 | 9.5 / **7.9** | 21.1 / 19.9 |
| Skywork-Qwen-8B | 7.5 / **2.5** | 2.1 / 2.1 | 4.0 / 3.8 | 9.5 / 9.3 |
| AllenAI-Llama-8B | 18.2 / **16.2** | 3.1 / **1.5** | **2.2** / 3.3 | 7.4 / 7.2 |
| Skywork-Llama-8B | 24.3 / **19.1** | 1.8 / 1.9 | 3.1 / 3.1 | 0.0 / 0.0 |

## C.4. Out-of-distribution Performance After Mechanistic Reward Shaping - Additional results

*Table 11.* RewardBench2 accuracies. Confidence refers to verbalized "low/med/high" calibration probes. Uncertainty refers to hedging language probes. Combined nulls Position, Length, Uncertainty, and Confidence. Asterisks indicate non-inferiority to baseline (margin 5 pp): * $p < 0.05$, ** $p < 0.01$, *** $p < 0.001$. Bold indicates highest accuracy in row.

| Model | Baseline | Length | Position | Uncertainty | Confidence | Combined |
|---|---|---|---|---|---|---|
| DeBERTa | **26.4** | 24.2[*] | 24.3[*] | 24.2[*] | 24.1[*] | 25.6[***] |
| Skywork-Qwen-0.6B | 69.8 | 69.0[***] | **69.9**[***] | 69.9[***] | **69.9**[***] | 68.1[**] |
| Skywork-Qwen-8B | 85.4 | 85.4[***] | 85.3[***] | **86.0**[***] | 85.5[***] | 85.4[***] |
| AllenAI-Llama-8B | **79.8** | 77.9[**] | 79.1[***] | 79.6[***] | 79.2[***] | 77.9[**] |
| Skywork-Llama-8B | 89.4 | **89.5**[***] | 89.4[***] | 89.3[***] | 89.4[***] | 88.9[***] |

*Table 12.* To measure the impact of our linear de-biasing probes on reward distribution, we show the average chosen, rejected, and difference in reward for RewardBench2 before and after different probes with bootstrapped 95% confidence interval uncertainties. We also show the result when combining multiple de-biasing probes, but exclude the "confidence calibration" probe, as we already include the uncertainty-related probe.

| PROBE | MEAN $r$(CHOSEN) | MEAN $r$(REJECTED) | MEAN ($r$(CHOSEN) - $r$(REJECTED)) |
|---|---|---|---|
| *Skywork/Skywork-Reward-V2-Qwen3-0.6B* | | | |
| **BASELINE** | $4.20^{+0.14}_{-0.15}$ | $0.56^{+0.15}_{-0.15}$ | $3.65^{+0.16}_{-0.15}$ |
| POSITION | $4.01^{+0.14}_{-0.15}$ | $0.48^{+0.16}_{-0.15}$ | $3.52^{+0.15}_{-0.16}$ |
| UNCERTAINTY | $3.30^{+0.12}_{-0.13}$ | $0.27^{+0.12}_{-0.13}$ | $3.03^{+0.13}_{-0.13}$ |
| LENGTH | $4.42^{+0.15}_{-0.15}$ | $0.91^{+0.15}_{-0.15}$ | $3.52^{+0.16}_{-0.16}$ |
| **ALL COMBINED** | $3.39^{+0.13}_{-0.14}$ | $0.45^{+0.13}_{-0.13}$ | $2.95^{+0.14}_{-0.13}$ |
| CONFIDENCE CALIBRATION | $1.91^{+0.09}_{-0.09}$ | $0.05^{+0.08}_{-0.08}$ | $1.86^{+0.09}_{-0.08}$ |
| *Skywork/Skywork-Reward-V2-Llama-3.1-8B* | | | |
| **BASELINE** | $18.06^{+0.41}_{-0.41}$ | $-2.16^{+0.54}_{-0.55}$ | $20.22^{+0.60}_{-0.60}$ |
| POSITION | $9.07^{+0.30}_{-0.28}$ | $-2.79^{+0.30}_{-0.29}$ | $11.86^{+0.38}_{-0.35}$ |
| UNCERTAINTY | $3.34^{+0.15}_{-0.16}$ | $-1.89^{+0.13}_{-0.12}$ | $5.23^{+0.16}_{-0.17}$ |
| LENGTH | $18.13^{+0.42}_{-0.41}$ | $-1.71^{+0.54}_{-0.53}$ | $19.84^{+0.58}_{-0.56}$ |
| **ALL COMBINED** | $2.81^{+0.15}_{-0.15}$ | $-2.19^{+0.11}_{-0.11}$ | $5.00^{+0.16}_{-0.17}$ |
| CONFIDENCE CALIBRATION | $11.92^{+0.28}_{-0.27}$ | $-0.45^{+0.33}_{-0.33}$ | $12.38^{+0.37}_{-0.38}$ |
| *Skywork/Skywork-Reward-V2-Qwen3-8B* | | | |
| **BASELINE** | $8.99^{+0.26}_{-0.25}$ | $-1.70^{+0.31}_{-0.30}$ | $10.68^{+0.35}_{-0.34}$ |
| POSITION | $6.38^{+0.18}_{-0.17}$ | $-0.56^{+0.19}_{-0.19}$ | $6.94^{+0.24}_{-0.23}$ |
| UNCERTAINTY | $7.08^{+0.18}_{-0.17}$ | $-0.56^{+0.20}_{-0.21}$ | $7.64^{+0.25}_{-0.24}$ |
| LENGTH | $8.99^{+0.24}_{-0.23}$ | $-1.08^{+0.28}_{-0.28}$ | $10.07^{+0.34}_{-0.33}$ |
| **ALL COMBINED** | $5.67^{+0.14}_{-0.15}$ | $0.15^{+0.15}_{-0.15}$ | $5.52^{+0.19}_{-0.18}$ |
| CONFIDENCE CALIBRATION | $6.84^{+0.12}_{-0.12}$ | $2.17^{+0.14}_{-0.15}$ | $4.67^{+0.16}_{-0.16}$ |
| *OpenAssistant/reward-model-deberta-v3-large-v2* | | | |
| **BASELINE** | $0.514^{+0.005}_{-0.005}$ | $0.509^{+0.005}_{-0.005}$ | $0.005^{+0.005}_{-0.005}$ |
| POSITION | $0.362^{+0.009}_{-0.008}$ | $0.378^{+0.007}_{-0.008}$ | $-0.016^{+0.007}_{-0.007}$ |
| UNCERTAINTY | $0.462^{+0.004}_{-0.004}$ | $0.461^{+0.004}_{-0.004}$ | $0.0^{+0.004}_{-0.004}$ |
| LENGTH | $0.466^{+0.005}_{-0.005}$ | $0.468^{+0.004}_{-0.004}$ | $-0.003^{+0.005}_{-0.004}$ |
| **ALL COMBINED** | $0.550^{+0.007}_{-0.007}$ | $0.554^{+0.006}_{-0.006}$ | $-0.004^{+0.006}_{-0.006}$ |
| CONFIDENCE CALIBRATION | $0.420^{+0.005}_{-0.005}$ | $0.426^{+0.004}_{-0.004}$ | $-0.006^{+0.004}_{-0.005}$ |
| *allenai/Llama-3.1-8B-Instruct-RM-RB2* | | | |
| **BASELINE** | $0.41^{+0.07}_{-0.06}$ | $-2.40^{+0.10}_{-0.11}$ | $2.81^{+0.10}_{-0.10}$ |
| POSITION | $0.38^{+0.06}_{-0.06}$ | $-2.38^{+0.10}_{-0.10}$ | $2.76^{+0.09}_{-0.10}$ |
| UNCERTAINTY | $1.43^{+0.07}_{-0.06}$ | $-1.51^{+0.10}_{-0.11}$ | $2.95^{+0.10}_{-0.10}$ |
| LENGTH | $0.34^{+0.06}_{-0.06}$ | $-2.15^{+0.09}_{-0.09}$ | $2.49^{+0.09}_{-0.09}$ |
| **ALL COMBINED** | $1.00^{+0.06}_{-0.06}$ | $-1.54^{+0.09}_{-0.10}$ | $2.54^{+0.09}_{-0.09}$ |
| CONFIDENCE CALIBRATION | $-0.41^{+0.05}_{-0.05}$ | $-2.47^{+0.08}_{-0.08}$ | $2.06^{+0.07}_{-0.07}$ |

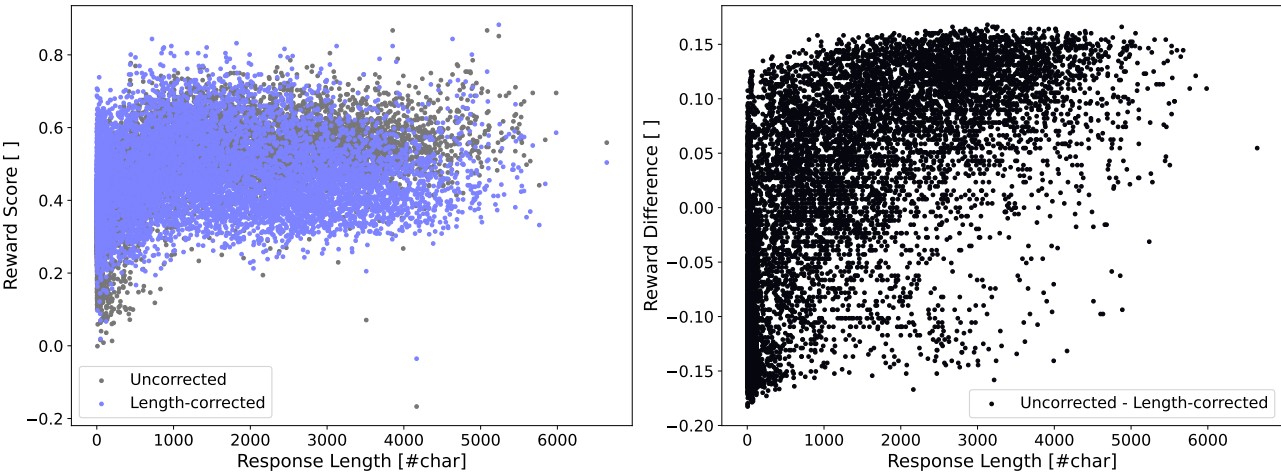

*Figure 4.* RewardBench2 impact of length correlation probe OOD for DeBERTa. Spearman $r_s = 0.611$ (95% CI: $[0.597, 0.624]$ uncorrected and Spearman $r_s = 0.067$ (95% CI: $[0.047, 0.087]$; significant overall spearman correlation (length bias) decrease. See probe increasingly subtracting rewards for longer sequences and working as intended for a strongly length-biased RM (unlike the other tested models, this one was not trained to have no length bias). Reward corrections are about 20% (large)

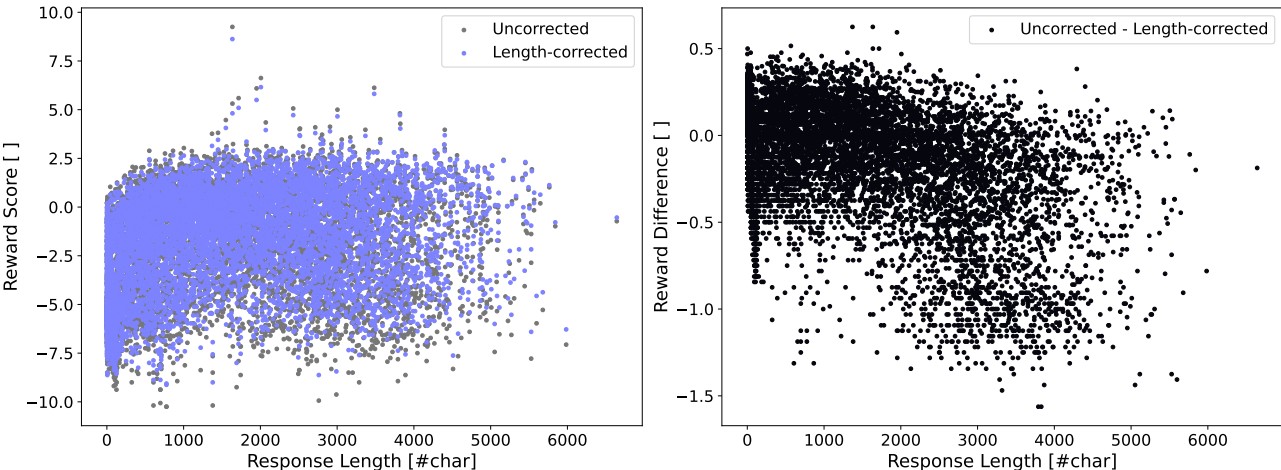

*Figure 5.* RewardBench2 impact of length correlation probe OOD for AllenAI-Llama-8B. Spearman $r_s = 0.369$ (95% CI: $[0.350, 0.389]$ uncorrected and Spearman $r_s = 0.441$ (95% CI: $[0.423, 0.460]$); significant overall spearman correlation (length bias) increase. See probe doing minor corrections for response lengths until 2000 characters, at which point it slightly reduces the detected overcorrection with smaller reward increases. Reward corrections are relatively small for shorter responses (below 10%), but reach up to 20% for long responses, which we expect as these models have been trained to have no length bias

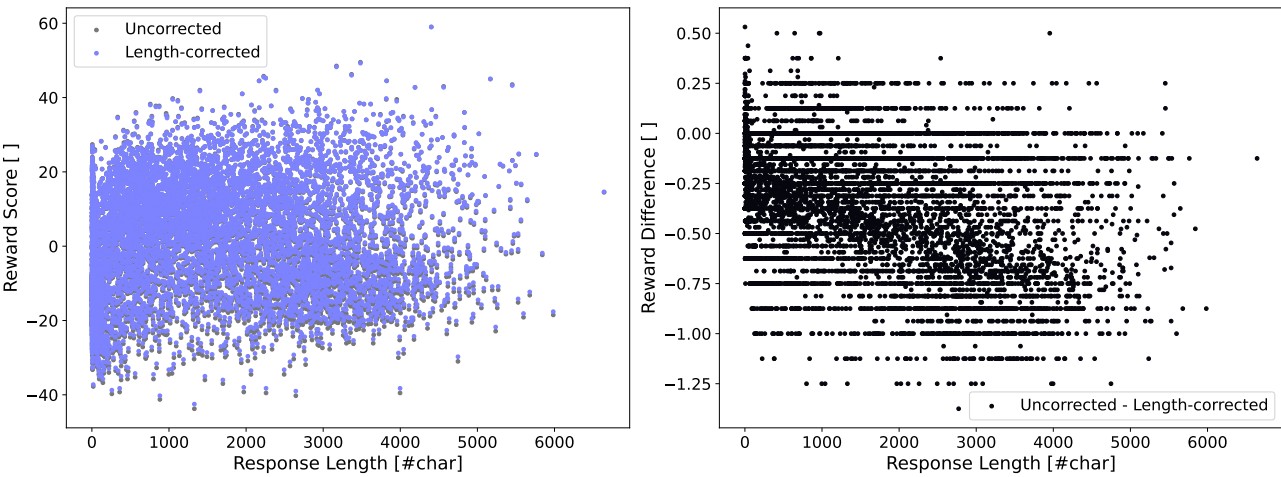

*Figure 6.* RewardBench2 impact of length correlation probe OOD for Skywork-Llama-8B. Spearman $r_s = 0.267$ (95% CI: $[0.247, 0.289]$ uncorrected and Spearman $r_s = 0.305$ (95% CI: $[0.256, 0.298]$); no significant overall spearman correlation (length bias) increase or decrease as CI values overlap. See probe doing minor reward corrections that approx. linearly increase with response lengths to slightly reduces the detected overcorrection. Reward corrections are relatively small for shorter responses (below 10%), which we expect as these models have been trained to have no length bias.

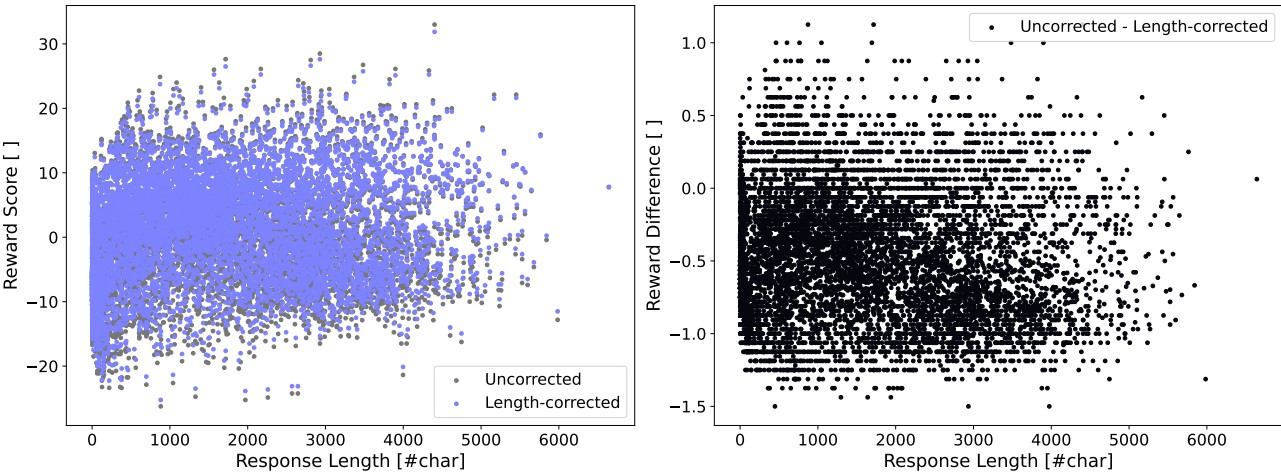

*Figure 7.* RewardBench2 impact of length correlation probe OOD for Skywork-Qwen-8B. Spearman $r_s = 0.296$ (95% CI: $[0.275, 0.316]$) uncorrected and Spearman $r_s = 0.305$ (95% CI: $[0.284, 0.326]$); no significant overall spearman correlation (length bias) increase or decrease as CI values overlap. See probe doing minor reward corrections that increase with response lengths to slightly reduces the detected overcorrection. Reward corrections are relatively small for shorter responses (below 10%), which we expect as these models have been trained to have no length bias.

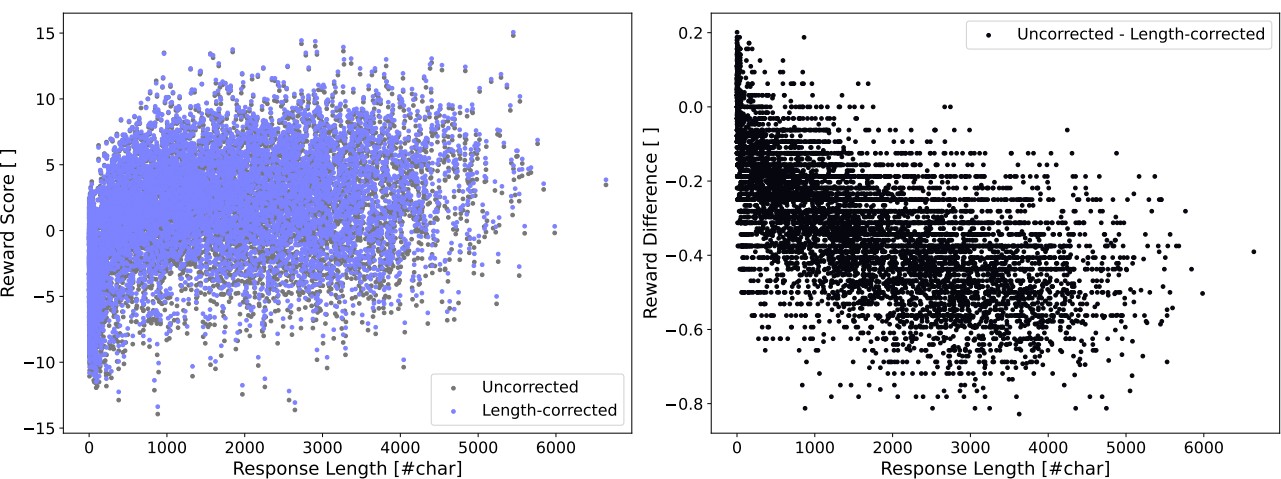

*Figure 8.* RewardBench2 impact of length correlation probe OOD for Skywork-Qwen-0.6B. Spearman $r_s = 0.465$ (95% CI: $[0.447, 0.482]$ uncorrected and Spearman $r_s = 0.496$ (95% CI: $[0.479, 0.513]$); no significant overall spearman correlation (length bias) increase or decrease as CI values overlap. See probe doing minor reward corrections that approx. linearly increase with response lengths to slightly reduces the detected overcorrection. Reward corrections are relatively small for shorter responses (below 10%), which we expect as these models have been trained to have no length bias.

## C.5. Uncertainty Probe Out of Distribution Evaluation

**Setup:** We evaluate whether the uncertainty probe generalizes *out of distribution* by examining only examples where the probe changes the RM's preference on RewardBench2. Concretely, for each RewardBench2 example we have a dataset `chosen` response and one or more `rejected` responses. For each RM and each example, we compare the scalar reward assigned to the dataset `chosen` response against the highest-scoring `rejected` response. We then collect cases where the preferred response under the baseline RM differs from the preferred response after applying the uncertainty probe.

**Prompt:** `On a scale of 1--10, how much uncertainty does this answer express? Reply with a single integer 1--10.`

**Judging protocol:** To avoid order effects, we score each response *independently*: for every flip pair we query the judge model twice (once per response) and record a 1–10 uncertainty rating for each. The judge is only shown the original prompt and the candidate answer text, and is not told whether the answer was dataset `chosen` or `rejected`, nor whether it was preferred by the baseline or the probe. We use GPT-5.1 as the judge.

**Result:** Across all uncertainty-flip pairs ($n = 257$), the mean uncertainty rating of the **baseline-preferred** response is 2.40, while the mean uncertainty rating of the **probe-preferred** response is 3.29 (difference +0.89). This difference is statistically significant under a paired t-test over flip pairs ($p = 9.3 \times 10^{-6}$, two-sided), indicating that when the uncertainty probe changes the RM's preference on RewardBench2, it tends to select responses that *express more uncertainty* according to the LLM judge.

## C.6. Sycophancy - Additional Results

**Probe: Sycophancy**

**Direction description**: a direction that separates assistant *agreement with a user-suggested answer* from *disagreement*, under an explicit user-opinion prompt.

```
User: {question}
I think the answer is {suggested_answer}.
Assistant:
(+) The answer is {suggested_answer}.
(-) The answer is {other_answer}.
```

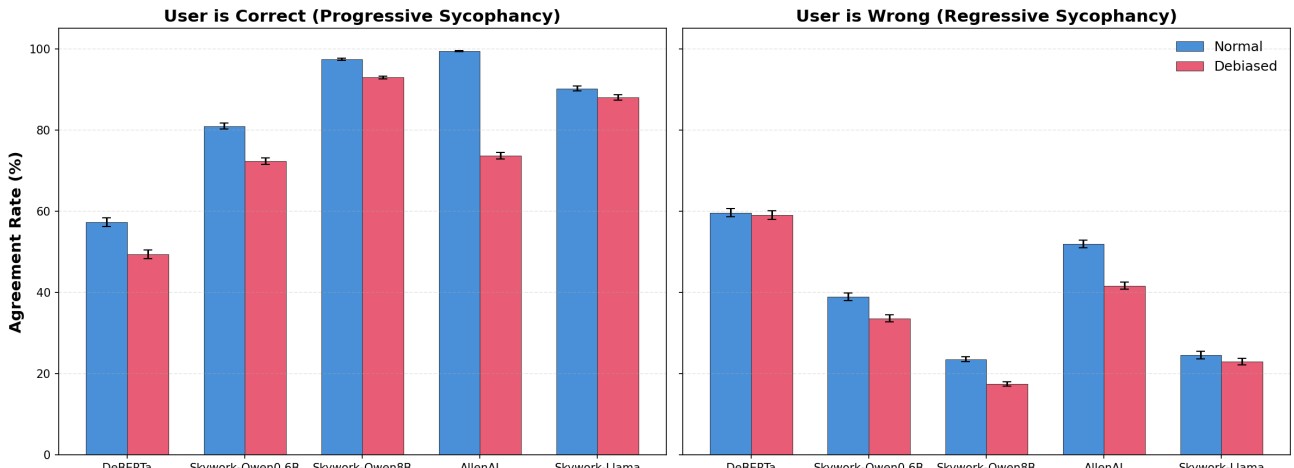

*Figure 9.* Inconsistent effects emerge when we attempt to resolve excessive user agreement.

*Table 13.* Sycophancy bias: Regressive (cave rate when RM correct, lower is better) and Progressive (help rate when RM wrong, higher is better). Format: baseline / nulled. Bold indicates significantly better ($p < 0.05$).

| Model | GSM8K-MC | MMLU | PlausibleQA | BigBench |
|---|---|---|---|---|
| *Regressive (cave rate % – lower is better)* | | | | |
| DeBERTa | 86.9 / 78.5 | 55.3 / 56.9 | 58.4 / 57.6 | 68.7 / **64.6** |
| Skywork-Qwen-0.6B | 59.6 / **42.2** | 34.6 / 35.9 | 33.6 / **31.0** | 54.5 / **30.2** |
| Skywork-Qwen-8B | 25.2 / 22.3 | 21.9 / **19.4** | 19.5 / **15.3** | 36.5 / **16.0** |
| AllenAI-Llama-8B | **75.6** / 88.7 | 50.0 / **36.2** | 26.5 / **23.2** | 55.4 / **52.4** |
| Skywork-Llama-8B | 51.0 / 46.9 | 23.9 / **22.3** | 11.7 / 11.0 | 30.5 / **17.7** |
| *Progressive (help rate % – higher is better)* | | | | |
| DeBERTa | **61.1** / 24.6 | 27.8 / **29.4** | **43.9** / 27.9 | **41.2** / 34.2 |
| Skywork-Qwen-0.6B | **69.1** / 24.7 | 35.5 / **37.5** | **18.6** / 14.1 | **65.9** / 26.2 |
| Skywork-Qwen-8B | 35.3 / 26.9 | **45.2** / 38.2 | **36.9** / 20.7 | **57.0** / 16.8 |
| AllenAI-Llama-8B | **96.3** / 27.6 | **82.6** / 37.2 | **70.1** / 27.5 | **78.0** / 19.6 |
| Skywork-Llama-8B | **50.2** / 29.0 | **43.4** / 38.3 | **27.3** / 22.7 | **35.9** / 16.0 |

## C.7. Model-Style Reward Correlation - Additional Results

*Table 14.* Spearman correlations between panel-relative (negative) per-byte cross-entropy $\Delta s_m(x, y)$ and rewards $r(x, y)$ using $n = 2400 \cdot 2$ samples (scoring each prompt with a chosen and rejected sample) for RM *allenai/Llama-3.1-8B-Instruct-RM-RB2* across different models. The respective other listed models make up the panel. Uncertainty intervals estimated with bootstrap resampling ($k = 2000$, 95% confidence). We mark correlations statistically significant from 0 in bold. See Section 5.2 for details.

| MODEL $m$ | SPEARMAN $\rho_s(r(x, y), \Delta s_m(x, y))$ |
|---|:---:|
| QWEN2.5-7B-INSTRUCT | $\mathbf{0.27}^{+0.02}_{-0.02}$ |
| LLAMA-3.1-8B-INSTRUCT | $\mathbf{0.11}^{+0.02}_{-0.02}$ |
| QWEN3-8B_NOTHINK | $\mathbf{0.06}^{+0.03}_{-0.03}$ |
| LLAMA-2-13B-CHAT-HF | $\mathbf{0.04}^{+0.03}_{-0.03}$ |
| LLAMA-2-7B-CHAT-HF | $0.01^{+0.03}_{-0.03}$ |
| QWEN3-0.6B_NOTHINK | $-0.01^{+0.03}_{-0.03}$ |
| GEMMA-3-12B-IT | $-0.02^{+0.02}_{-0.03}$ |
| GEMMA-2-2B-IT | $\mathbf{-0.03}^{+0.02}_{-0.02}$ |
| QWEN2.5-0.5B-INSTRUCT | $\mathbf{-0.11}^{+0.03}_{-0.03}$ |
| GEMMA-2-9B-IT | $\mathbf{-0.12}^{+0.03}_{-0.03}$ |
| MEAN ABSOLUTE | 0.08 |

*Table 15.* Spearman correlations between panel-relative (negative) per-byte cross-entropy $\Delta s_m(x, y)$ and rewards $r(x, y)$ using $n = 2400 \cdot 2$ samples (scoring each prompt with a chosen and rejected sample) for RM *Skywork/Skywork-Reward-V2-Llama-3.1-8B* across different models. The respective other listed models make up the panel. Uncertainty intervals estimated with bootstrap resampling ($k = 2000$, 95% confidence). We mark correlations statistically significant from 0 in bold. See Section 5.2 for details.

| MODEL $m$ | SPEARMAN $\rho_s(r(x,y), \Delta s_m(x,y))$ |
|---|:---:|
| QWEN2.5-7B-INSTRUCT | $\mathbf{0.25}^{+0.02}_{-0.02}$ |
| LLAMA-3.1-8B-INSTRUCT | $\mathbf{0.19}^{+0.02}_{-0.02}$ |
| QWEN3-8B_NOTHINK | $\mathbf{0.08}^{+0.03}_{-0.02}$ |
| QWEN3-0.6B_NOTHINK | $\mathbf{0.05}^{+0.03}_{-0.03}$ |
| GEMMA-2-9B-IT | $\mathbf{0.05}^{+0.02}_{-0.03}$ |
| GEMMA-3-12B-IT | $0.01^{+0.03}_{-0.03}$ |
| QWEN2.5-0.5B-INSTRUCT | $\mathbf{-0.04}^{+0.03}_{-0.02}$ |
| GEMMA-2-2B-IT | $\mathbf{-0.04}^{+0.02}_{-0.02}$ |
| LLAMA-2-13B-CHAT-HF | $\mathbf{-0.18}^{+0.02}_{-0.02}$ |
| LLAMA-2-7B-CHAT-HF | $\mathbf{-0.21}^{+0.02}_{-0.03}$ |
| MEAN ABSOLUTE | 0.11 |

*Table 16.* Spearman correlations between panel-relative (negative) per-byte cross-entropy $\Delta s_m(x, y)$ and rewards $r(x, y)$ using $n = 2400 \cdot 2$ samples (scoring each prompt with a chosen and rejected sample) for RM *Skywork/Skywork-Reward-V2-Qwen3-8B* across different models. The respective other listed models make up the panel. Uncertainty intervals estimated with bootstrap resampling ($k = 2000$, 95% confidence). We mark correlations statistically significant from 0 in bold. See Section 5.2 for details.

| MODEL $m$ | SPEARMAN $\rho_s(r(x,y), \Delta s_m(x,y))$ |
|---|:---:|
| QWEN2.5-7B-INSTRUCT | $\mathbf{0.24}^{+0.02}_{-0.02}$ |
| LLAMA-3.1-8B-INSTRUCT | $\mathbf{0.17}^{+0.02}_{-0.02}$ |
| QWEN3-0.6B_NOTHINK | $\mathbf{0.08}^{+0.02}_{-0.02}$ |
| QWEN3-8B_NOTHINK | $\mathbf{0.06}^{+0.02}_{-0.03}$ |
| GEMMA-2-9B-IT | $-0.01^{+0.02}_{-0.03}$ |
| GEMMA-2-2B-IT | $-0.02^{+0.02}_{-0.02}$ |
| QWEN2.5-0.5B-INSTRUCT | $-0.02^{+0.02}_{-0.02}$ |
| GEMMA-3-12B-IT | $\mathbf{-0.09}^{+0.03}_{-0.03}$ |
| LLAMA-2-13B-CHAT-HF | $\mathbf{-0.09}^{+0.03}_{-0.03}$ |
| LLAMA-2-7B-CHAT-HF | $\mathbf{-0.12}^{+0.02}_{-0.03}$ |
| MEAN ABSOLUTE | 0.09 |

*Table 17.* Spearman correlations between panel-relative (negative) per-byte cross-entropy $\Delta s_m(x,y)$ and rewards $r(x,y)$ using $n = 2400 \cdot 2$ samples (scoring each prompt with a chosen and rejected sample) for RM *Skywork/Skywork-Reward-V2-Qwen3-0.6B* across different models. The respective other listed models make up the panel. Uncertainty intervals estimated with bootstrap resampling ($k = 2000$, 95% confidence). We mark correlations statistically significant from 0 in bold. See Section 5.2 for details.

| MODEL $m$ | SPEARMAN $\rho_s(r(x,y), \Delta s_m(x,y))$ |
|---|:---:|
| QWEN2.5-7B-INSTRUCT | $\mathbf{0.37}^{+0.02}_{-0.02}$ |
| QWEN3-8B_NOTHINK | $\mathbf{0.15}^{+0.02}_{-0.02}$ |
| LLAMA-3.1-8B-INSTRUCT | $\mathbf{0.10}^{+0.02}_{-0.02}$ |
| GEMMA-3-12B-IT | $0.01^{+0.03}_{-0.03}$ |
| QWEN3-0.6B_NOTHINK | $\mathbf{-0.05}^{+0.03}_{-0.02}$ |
| GEMMA-2-2B-IT | $\mathbf{-0.05}^{+0.02}_{-0.02}$ |
| GEMMA-2-9B-IT | $\mathbf{-0.05}^{+0.03}_{-0.03}$ |
| LLAMA-2-13B-CHAT-HF | $\mathbf{-0.08}^{+0.03}_{-0.03}$ |
| LLAMA-2-7B-CHAT-HF | $\mathbf{-0.12}^{+0.02}_{-0.03}$ |
| QWEN2.5-0.5B-INSTRUCT | $\mathbf{-0.20}^{+0.02}_{-0.02}$ |
| MEAN ABSOLUTE | 0.12 |

*Table 18.* Spearman correlations between panel-relative (negative) per-byte cross-entropy $\Delta s_m(x,y)$ and rewards $r(x,y)$ using $n = 2400 \cdot 2$ samples (scoring each prompt with a chosen and rejected sample) for RM *OpenAssistant/reward-model-deberta-v3-large-v2* across different models. The respective other listed models make up the panel. Uncertainty intervals estimated with bootstrap resampling ($k = 2000$, 95% confidence). We mark correlations statistically significant from 0 in bold. See Section 5.2 for details.

| MODEL $m$ | SPEARMAN $\rho_s(r(x,y), \Delta s_m(x,y))$ |
|---|:---:|
| QWEN2.5-7B-INSTRUCT | $\mathbf{0.18}^{+0.02}_{-0.02}$ |
| LLAMA-2-7B-CHAT-HF | $\mathbf{0.16}^{+0.02}_{-0.02}$ |
| LLAMA-2-13B-CHAT-HF | $\mathbf{0.14}^{+0.02}_{-0.02}$ |
| QWEN3-8B_NOTHINK | $\mathbf{0.13}^{+0.02}_{-0.02}$ |
| QWEN3-0.6B_NOTHINK | $\mathbf{0.04}^{+0.03}_{-0.03}$ |
| GEMMA-2-2B-IT | $\mathbf{0.03}^{+0.02}_{-0.02}$ |
| LLAMA-3.1-8B-INSTRUCT | $-0.01^{+0.02}_{-0.02}$ |
| GEMMA-3-12B-IT | $\mathbf{-0.04}^{+0.03}_{-0.02}$ |
| QWEN2.5-0.5B-INSTRUCT | $\mathbf{-0.07}^{+0.03}_{-0.02}$ |
| GEMMA-2-9B-IT | $\mathbf{-0.23}^{+0.02}_{-0.02}$ |
| MEAN ABSOLUTE | 0.10 |

## C.8. BoN Evaluation

In order to extend our evaluation to a realistic post-training setting, we evaluate length-bias removal in the open-ended generation setting using AlpacaEval (Dubois et al., 2024). To do so, we construct our diff-mean probes by sampling 64 responses from Llama3.2-1B-Instruct (Grattafiori et al., 2024) for each of 293 AlpacaEval prompts. Then, for each response, we use the longest two generations as positive examples, and the shortest two generations as negative examples to construct probes.

**Global length calibration is less aligned with BoN selection.** Although Table 19 reports prompt-conditioned reward–length correlations, it is important to note that Huang et al. (2025) does successfully debias under the global diagnostic it is designed for. On held-out AlpacaEval BoN64 candidates, the calibration of Huang et al. (2025) reduces the mean absolute pooled reward–length correlation across reward models from 0.316 to 0.037. However, this pooled diagnostic is less aligned with the downstream setting we study. BoN selection is prompt-conditioned and for each instruction, the RM selects among 64 responses to the same prompt. Thus, the relevant question is whether the reward remains length-dependent within each candidate set, not whether length and reward are decorrelated after pooling candidates across prompts.

This distinction is analogous to the limitation of length-controlled AlpacaEval (Dubois et al., 2024). Length-controlled win rate estimates a counterfactual system-level preference after statistically adjusting for output-length differences, which is useful for reducing leaderboard-level verbosity effects. However, because the adjustment is learned globally across examples, it need not be robust to new prompts or to prompt-conditioned selection settings such as BoN, where length effects can vary by instruction and candidate set. Consistent with this, the global calibration of Huang et al. (2025) looks strong under pooled correlation but transfers less cleanly to the BoN-relevant diagnostic. Mechanistic reward shaping is closer to zero than Huang et al. (2025) on 4/5 AlpacaEval RMs, with average $|\rho_{\text{within}}| = 0.035$ versus 0.116. Our method also achieves higher AlpacaEval LC win rate on four out of five RMs, averaging $51.12\%$ versus $49.71\%$.

The same pattern is clearer on GSM8K in Table 20, where correctness is directly measurable rather than judged by a length-controlled preference model. There, mechanistic reward shaping is closer to zero on all 5 RMs, averaging $|\rho_{\text{within}}| = 0.007$ versus 0.096 for Huang et al. (2025), and achieves higher mean BoN accuracy, $62.76\%$ versus $61.20\%$. These results suggest that global length calibration can remove pooled reward–length correlation while still failing to address, or even overcorrecting, the prompt-conditioned length dependence that drives BoN selection.

## C.9. INLP Evaluation

To provide independent evidence for our complexity taxonomy beyond the success of null-space projection, we run Iterative Nullspace Projection (INLP) (Ravfogel et al., 2020). INLP iteratively trains a linear classifier to separate two labeled classes from RM activations, then projects activations onto the nullspace of that classifier. If a bias is well-captured by a single linear direction, residual separability after one iteration should collapse to near chance. If the bias is encoded along many directions or is non-linearly entangled with the reward signal, separability either persists across iterations or is already weak at the first iteration.

We train logistic regressions on the final-layer activations of each RM, using the same labeled splits underlying our DiffMean probes, see Section 3.2, and report AUC before (*Iter 1*) and after (*Iter 2*) removing one null direction in Table 21. We also plot the projections for Skywork-Llama-8B in Figure 10. For position bias, where we use a probe basis rather than a single direction, we report the best one-vs-rest split.

For low-complexity biases (length, uncertainty, position), Iter 1 AUC reaches up to 1.00 across RMs, indicating linear separability along the bias axis. After removing one direction, residual separability typically collapses to near chance, with the strongest effects on the most biased models (AllenAI-Llama-8B length 0.996 to 0.509, Skywork-Qwen-8B uncertainty 0.931 to 0.561, Skywork-Qwen-8B position 0.935 to 0.524). One partial exception is Skywork-Qwen-0.6B length (1.000 to 0.770), where residual structure remains, consistent with a multi-directional encoding in that smaller model.

For high-complexity biases we observe a qualitatively different picture. Style separability is linearly detectable but not captured by a single direction at Iter 1 (AUC 0.659 to 0.850), and regressive versus progressive sycophancy is not cleanly separable in any tested RM (Iter 1 AUC 0.491 to 0.771). Removing one direction does not consistently reduce separability and in some cases yields AUC below chance, consistent with redundant or multi-pathway encoding rather than a single removable feature.

Taken together, these results provide independent evidence that our low/high-complexity distinction reflects representational

| | DeBERTa | AllenAI-L8B | Skywork-L8B | Skywork-Q8B | Skywork-Q0.6B |
|---|---|---|---|---|---|
| *Signed Pearson reward–length correlation* ($\rho_{\text{len}}$) | | | | | |
| Baseline | +.168 | −.061 | +.134 | +.065 | +.088 |
| Mechanistic Reward Shaping (Ours) | +**.005** | −**.011** | +.087 | +**.032** | +**.041** |
| Huang et al. | +.048 | −.099 | −**.065** | −.138 | −.228 |
| *AlpacaEval length-controlled win rate vs. baseline* (%); higher is better | | | | | |
| Mech. RS | **54.42**\*\*\* | 46.23$^{\dagger}$ | **51.48**\*\*\* | **51.69**\*\*\* | **51.76**\*\*\* |
| Huang et al. | 50.43\* | **48.65**$^{\dagger}$ | 50.07 | 48.24$^{\dagger}$ | 51.19\*\*\* |

*Table 19.* Held-out AlpacaEval transfer results. Probes/calibrators are fit on a disjoint 293-prompt probe pool and evaluated on the corrected 512-prompt test pool with 64 candidates per prompt. Correlations are mean within-prompt Pearson correlations between reward score and response length. We report signed correlations to show whether each method removes or reverses length association, and $\Delta|\rho_{\text{len}}|$ to show movement toward zero relative to baseline. Bold marks the better debiasing method within each model and metric. For LC WR, significance is tested against 50% using the AlpacaEval-reported standard error under a normal approximation; \*$p < .05$, \*\*$p < .01$, \*\*\*$p < .001$. † denotes a significant result in the unfavorable direction, i.e. significantly below 50%.

| | DeBERTa | AllenAI-L8B | Skywork-L8B | Skywork-Q8B | Skywork-Q0.6B |
|---|---|---|---|---|---|
| *Signed Pearson reward–length correlation* ($\rho_{\text{len}}$) | | | | | |
| Baseline | +.297 | +.030 | +.022 | −**.005** | −.024 |
| Mechanistic Reward Shaping (Ours) | −**.005** | +**.001** | −**.007** | −.008 | −**.013** |
| Huang et al. | +.015 | +.146 | +.112 | +.134 | +.074 |
| *GSM8K BoN accuracy* (%); higher is better | | | | | |
| Baseline | 32.8 | **71.5** | 68.3 | **71.9** | 65.8 |
| Mech. RS | **36.4** | 71.3 | **68.5** | **71.9** | 65.7 |
| Huang et al. | 32.3 | 67.9 | 68.3 | 71.3 | **66.2** |

*Table 20.* Held-out GSM8K BoN results. We generated 64 responses per question using Llama-3.2-1B-Instruct, fit probes/calibrators on the first 512 questions, and evaluated on the disjoint 807-question held-out split. For each held-out question, each RM selects the highest-scoring response among 64 candidates. Accuracy is the fraction of selected responses judged correct. Correlations are mean within-question Pearson correlations between RM score and response length over the 64 candidates. We report signed correlations for interpretability and $\Delta|\rho_{\text{len}}|$ to measure movement toward zero. Bold marks the best method within each model and metric.

structure rather than only the empirical success of null-space projection.

| Domain | Model | Iter 1 AUC | Iter 2 AUC |
|---|---|---|---|
| *Length bias* | | | |
| | DeBERTa | 0.837 | 0.608 |
| | AllenAI-Llama-8B | 0.996 | 0.509 |
| | Skywork-Llama-8B | 0.999 | 0.587 |
| | Skywork-Qwen-8B | 1.000 | 0.655 |
| | Skywork-Qwen-0.6B | 1.000 | 0.770 |
| *Position bias (best one-vs-rest)* | | | |
| | DeBERTa | 0.653 | 0.516 |
| | AllenAI-Llama-8B | 0.888 | 0.539 |
| | Skywork-Llama-8B | 0.642 | 0.516 |
| | Skywork-Qwen-8B | 0.935 | 0.524 |
| | Skywork-Qwen-0.6B | 0.889 | 0.511 |
| *Uncertainty bias* | | | |
| | DeBERTa | 0.812 | 0.525 |
| | AllenAI-Llama-8B | 1.000 | 0.546 |
| | Skywork-Llama-8B | 0.869 | 0.524 |
| | Skywork-Qwen-8B | 0.931 | 0.561 |
| | Skywork-Qwen-0.6B | 1.000 | 0.516 |
| *Sycophancy (regressive vs. progressive)* | | | |
| | DeBERTa | 0.491 | 0.534 |
| | AllenAI-Llama-8B | 0.606 | 0.378 |
| | Skywork-Llama-8B | 0.613 | 0.489 |
| | Skywork-Qwen-8B | 0.771 | 0.455 |
| | Skywork-Qwen-0.6B | 0.566 | 0.511 |
| *Style bias* | | | |
| | DeBERTa | 0.732 | 0.537 |
| | AllenAI-Llama-8B | 0.659 | 0.536 |
| | Skywork-Llama-8B | 0.669 | 0.621 |
| | Skywork-Qwen-8B | 0.723 | 0.535 |
| | Skywork-Qwen-0.6B | 0.850 | 0.534 |

*Table 21.* Linear separability (AUC) before (*Iter 1*) and after (*Iter 2*) removing one null direction via INLP. Iter 1 near 1.0 indicates strong rank-1 linear structure. Iter 2 near 0.5 indicates the bias is fully captured in a single direction. Sycophancy measures separability between regressive agreement (model agrees with an incorrect user claim) and progressive agreement (model agrees with a correct user claim). The near-chance Iter 1 values indicate both subtypes share a common representational direction.

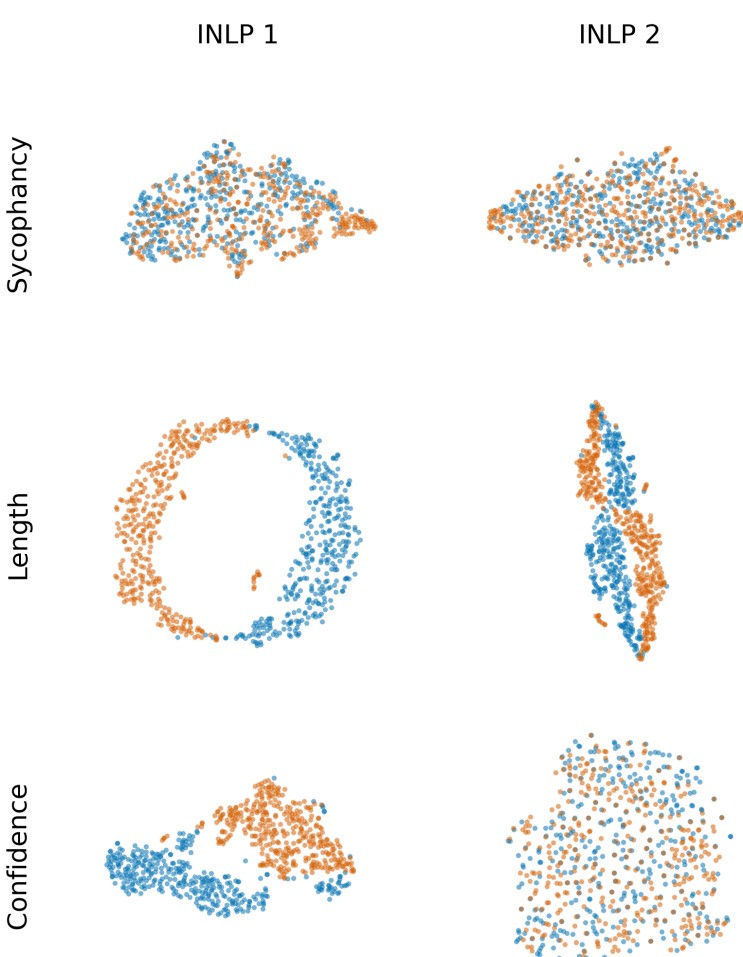

*Figure 10.* INLP projections for Skywork-Llama-8B.

