# OpenReview forum: "One Bias After Another: Mechanistic Reward Shaping and Persistent Biases in Language Reward Models"
_ICML.cc/2026/Conference — ICML 2026 regular_

### Official Review · Reviewer_UW6C · 2026-03-06

**Soundness:** 3
**Presentation:** 3
**Significance:** 3
**Originality:** 3
**Overall Recommendation:** 4
**Confidence:** 4

**Summary:**

This paper systematically audits five contemporary reward models (RMs) for biases along five dimensions (length, uncertainty, position, sycophancy, and model-style sensitivity) and categorizes them into low-complexity biases amenable to linear intervention and high-complexity biases resistant to it. For the low-complexity category, the authors propose a post-hoc, inference-time intervention that identifies bias-encoding directions in the RM's final-layer activations using difference-of-means probes and removes them via null-space projection. Evaluated on static benchmarks including RewardBench-2, the method reduces targeted biases without statistically significant degradation to ranking accuracy. Two novel RM biases are reported: position bias and model-style reward sensitivity. The central question of whether these biases can be mitigated cheaply and effectively is partially answered: the intervention works on the subset of biases the authors classify as low-complexity, but the paper does not validate whether this translates to improved outcomes in downstream policy optimization.

**Compliance With Llm Reviewing Policy:**

Affirmed.

**Final Justification:**

Authors addressed my core concerns, with especially end-to-end RLHF experiments. I updated my overall score to 4, and soundness to 3.

**Key Questions For Authors:**

How do you rule out the possibility that the linear probe captures a low-dimensional projection of a more complex underlying geometry, rather than the bias being strictly linear? What evidence would change your taxonomy in Section 3.1?

Can you provide an end-to-end RLHF training run, even at small scale, to demonstrate that the debiased RM actually prevents reward hacking and produces a less biased downstream policy?

Why was the null-space projection not benchmarked against existing inference-time baselines, such as length penalties or reward ensembles?

Section D.3 of the appendix contains what appears to be a leftover placeholder note: '[Need a bit more details on dataset creation, in particular that randomization is balanced etc.]'. Can you confirm whether position randomization was balanced across conditions in the position bias experiments, and provide the missing methodological details?

**Limitations:**

The authors do not adequately address the most critical limitation of their work. While they briefly note they do not examine how biases propagate through RLHF, they do not explicitly state that their proposed mitigation has never been evaluated in an actual training pipeline. A method motivated by preventing reward hacking must be tested in the environment where reward hacking occurs. This limitation should be prominently stated. Additionally, the epistemological limitation that successful linear intervention does not validate a linear representational geometry should be explicitly discussed rather than left implicit.

**Strengths And Weaknesses:**

Strengths
Practical utility. The proposed intervention is computationally cheap, data-efficient, and operates at inference time without retraining. For practitioners working with already-deployed RMs, this offers a directly applicable tool, which is a meaningful contribution given that most existing RM debiasing methods require expensive retraining or preference data curation.

Novel bias identification. Discovering position bias and continuous model-style reward sensitivity specifically within RMs is a valuable and original contribution. The model-style analysis, which introduces a panel-relative cross-entropy measure to isolate style from content, is methodologically careful.

Rigorous static evaluation. The use of out-of-distribution testing on RewardBench-2 with formal non-inferiority testing is a strong empirical design choice. The paper provides extensive per-model, per-dataset breakdowns with bootstrapped confidence intervals, which allows readers to assess the consistency of the results rather than relying on aggregated summaries.

Weaknesses
Circularity in the complexity taxonomy (Section 3.1). The paper's central organizing framework, classifying biases as "low-complexity" or "high-complexity," is defined by whether a linear null-space projection successfully mitigates them. This is circular: the success of the intervention becomes the evidence for the underlying representational claim. In high-dimensional transformer latent spaces, a linear probe can capture a low-dimensional projection of a much more complex, possibly non-linear geometry. The fact that a model maintains baseline performance after ablation does not prove the original feature was inherently linear; it may mean the model routes around the removed direction using redundant representations. The paper would benefit from either providing independent evidence for the linear geometry claim (e.g., through visualization of activation subspaces or comparison with non-linear probes) or reframing the taxonomy as empirically descriptive rather than mechanistically explanatory.

No end-to-end RLHF evaluation. The paper's stated motivation is preventing reward hacking during policy optimization, yet no language model policy is trained at any point. The evaluation remains entirely within static RM benchmarks. Without running an actual RLHF (e.g., PPO) or DPO loop with a debiased RM, it is unknown whether the intervention prevents reward hacking in a dynamic training environment, or whether the policy simply exploits the modified reward landscape through alternative pathways. This is not a minor omission but rather a fundamental gap between the paper's central motivation and its experimental evidence. Even a small-scale experiment with a 1B to 3B parameter policy trained with and without the debiased RM would substantially strengthen the claims.

No comparison with existing baselines. The null-space projection is not compared against standard inference-time mitigation techniques, such as simple mathematical length penalties applied to the reward score, reward ensembles, or other post-hoc reward shaping methods discussed in the related work. Without such comparisons, it is difficult to assess the relative merit of the proposed approach.

Presentation overstates mechanistic claims. The narrative framing presents the intervention as "mechanistic reward shaping" and ties it closely to the Linear Representation Hypothesis. However, what the method empirically demonstrates is a pragmatic and effective inference-time regularization technique. The gap between the mechanistic language and the actual evidence weakens the paper's credibility. A more measured framing that acknowledges the method as an engineering tool whose success is consistent with, but does not prove, linear geometry would be more appropriate.

---

> ### Author Rebuttal · Authors · 2026-03-31
>
> We thank Reviewer UW6C for detailed and thoughtful comments, and appreciate their acknowledgement of the rigor, novelty, and practical utility of our work. Below we address concerns raised:
>
> **Removing any Circularity in Complexity Taxonomy**
>
> We provide independent evidence exploring linearity of biases via Iterative Nullspace Projection (INLP; Ravfogel et al., 2020), which tests whether bias signals remain linearly decodable after removing the top direction. For low-complexity biases (length, uncertainty, calibration), we observe strong initial separability (AUCs up to 1.00), but after removing the top direction, residual separability collapses to near chance (AUC ≈ 0.50–0.57). This indicates that the signal is largely captured by a single dominant linear component. This effect is more pronounced in more biased models (e.g., Allen RM: length 0.996 → 0.58). For structured multi-probe biases such as position, we find a small low-rank subspace of collected probes suffices, and that the next plausible subspace cannot separate positions (macro AUC 0.88 → 0.51), indicating limited but multi-directional structure. In contrast, for higher-complexity biases, we don’t observe strong linear separability even at the first iteration (model-style signal AUC ≈ 0.58, sycophancy AUC ≈ 0.68), suggesting these effects are not captured by a single linear direction.
> These results strengthen our causal claim. Still, we will revise the paper to clarify that our taxonomy is primarily empirical and descriptive, supported by this independent analysis.
>
> **Demonstrating performance compared to prior work baselines**
>
> We compare our method to Huang et al. (2025), the only post-hoc calibration method that doesn't rely on additional models or extensive RM-retraining. Across RMs, we find that our method more effectively removes verbosity bias. On GSM8K with the Allen RM, the preference rate for longer correct responses increases from 56.4% (baseline) and 65.1% (Huang et al.) to 74.5% with our method. For DeBERTa, which exhibits strong length bias (82.1%), Huang et al. partially reduces this effect (55.5%), while our method removes it entirely (46.1%). Baseline performance for models without length variation are 47.5% for DeBERTa and 80.9% for the AllenRM, so our model more successfully minimizes the effect of verbosity on true accuracy.
> Zeyu Huang et al.. Post-hoc Reward Calibration: A Case Study on Length Bias. ICLR 2025.
>
> **Providing evidence of downstream RL impact**
>
> To show that our methods generalize to downstream RL, we conduct new best-of-N and PPO experiments using a new length probe constructed with the longest and shortest sampled responses for 293 AlpacaEval prompts. For 512 held out prompts, as N increases, baseline RMs amplify their inherent verbosity biases (DeBERTa longer, Allen shorter), while our debiased models attenuate or reverse these trends in agreement with our base paper findings. Then, we select the best-of-32 responses to 512 held out prompts, and find the DeBERTa baseline produces significantly longer responses than its debiased counterpart (+10 words), while the debiased Allen RM produces substantially longer responses than baseline (+24 words), correcting each model’s original bias direction. Under a gpt-5 AlpacaEval judge, the debiased DeBERTa RM achieves a higher win rate (17.19% vs 13.28%), and the debiased Allen RM achieves a similar win rate (2.73% vs 3.71%). This adds to our OOD RewardBench2 evidence , and supports that our method drives optimization toward debiased choices.
>
> We also trained Llama-3.2-1B PPO policies (512 steps) with baseline and debiased RMs, generating on 1.5k held-out prompts. For Allen, whose base RM penalizes verbosity by 25pp in controlled evaluation (Section 4.1), the debiased policy produced significantly longer responses (+2.14 words, p=0.03) and was preferred by a GPT-5 judge in 52.3% of non-ties, consistent with end-to-end correction of overcorrection bias. For DeBERTa, differences were not significant in either length or judge preference. We note that prior work demonstrating reward hacking via PPO relies on intentionally misspecified RMs with large bias magnitudes, while our intervention targets subtle biases in production-grade RMs.
>
> **In support of the Mechanistic Framing**
>
> We agree "mechanistic" warrants clarification. Saphra & Wiegreffe (2024) distinguish multiple valid uses of the term, and we situate ourselves at the model-internal causal intervention end of that spectrum, as opposed to full circuit-level explanation. Our method falls within this spectrum by identifying directions in the RM's activation space and performing causal interventions (null-space projection) that measurably change downstream reward behavior (causal rather than correlational). We will revise for clarity.
>
> Saphra and Wiegreffe. Mechanistic? BlackboxNLP Workshop 2024.
>
> **Leftover placeholder** was an oversight and will be removed (thank you!). The submission results were already complete.

---

> > ### Author Rebuttal · Reviewer_UW6C · 2026-03-31
> >
> > I thank the authors for their rebuttal. You directly addressed my core concerns, particularly by running the end-to-end RLHF experiments and providing independent evidence (INLP) for the complexity taxonomy. These additions successfully resolve my core concerns. I am raising my score to a 4.

---

### Official Review · Reviewer_DDEb · 2026-03-08

**Soundness:** 3
**Presentation:** 3
**Significance:** 3
**Originality:** 3
**Overall Recommendation:** 4
**Confidence:** 3

**Summary:**

This paper systematically measures biases in five reward models (RMs) used in RLHF, categorizes them into low-complexity (length, uncertainty, position) and high-complexity (sycophancy, model-style sensitivity) classes, and proposes a lightweight post-hoc debiasing method via linear activation probe null-space projection. The method mitigates low-complexity biases without degrading RewardBench-2 performance, while high-complexity biases are shown to resist linear interventions.

**Compliance With Llm Reviewing Policy:**

Affirmed.

**Final Justification:**

The rebuttal adequately addressed my concerns. I am maintaining my positive score.

**Key Questions For Authors:**

1. Can the length debiasing probe be evaluated on more datasets on other domains?
2. Are there any potential sources for the high-complexity biases?

**Limitations:**

yes

**Strengths And Weaknesses:**

**Strengths:**

- The research problem is important. Systematic evaluation of RM biases across multiple contemporary models, including the current SOTA, is a timely and valuable empirical contribution.
- The proposed method is technically sound, lightweight, and requires no retraining. The DiffMean + null-space projection approach is well-motivated by the linear representation hypothesis and generalizes OOD.

**Weaknesses:**

- The length debiasing probe is constructed and evaluated only on GSM8K. It is unclear whether the identified direction is a general length feature or a math-specific artifact. Evaluation on at least one additional domain (e.g., code or dialogue) is needed to support the paper's broader claims.
- For high-complexity biases (sycophancy, model-style sensitivity), the paper only observes and reports the phenomena without investigating their sources. Understanding why these biases arise would substantially strengthen the paper's significance beyond a descriptive study.

---

> ### Author Rebuttal · Authors · 2026-03-31
>
> We thank Reviewer DDEb for recognizing the importance and relevance of our work as well as our systematic approach, sound and lightweight methods.
>
> **Does the length probe generalize OOD?**
>
> We present OOD generalization results in Appendix Section D.4, where the probe trained on GSM8K synthetic data transfers to human-written RewardBench-2 responses, measurably changing reward-length correlations in the expected direction for each model.
>
> We also demonstrate generalization out-of-distribution by evaluating reward models under Best-of-N (BoN) selection on AlpacaEval prompts. As N increases, baseline reward models amplify their inherent verbosity biases (DeBERTa longer, Allen shorter), while our debiased models attenuate or reverse these trends in agreement with our base paper findings. At N=32, the DeBERTa baseline produces substantially longer outputs than its debiased counterpart ( $\sim$ 10 words longer, on average), while the debiased Allen RM produces substantially longer outputs than baseline ($\sim$ 24 words, on average), correcting each model’s bias direction. Under a GPT-5 AlpacaEval judge, this translates to downstream outcomes: the debiased DeBERTa RM achieves a higher win rate (17.19% vs 13.28%), while the debiased Allen RM achieves a comparable win rate (2.73% vs 3.71%) despite removing its bias toward shorter responses. This demonstrates that reward model biases materially affect selection under optimization, and that our method mitigates these effects.
>
> We agree that a more diverse probe construction would be warranted in many cases, even when biases are perfectly linear, in order to capture the correct direction by which each bias is mediated. Constructing and validating accurate probes is an active area of research that we see as synergistic and complementary to this work.
>
> **Are there any potential sources for the high-complexity biases?**
>
> While identifying the sources of high-complexity biases is beyond the main scope of this work, we agree this is an important question and will expand the discussion in a final version. Our results suggest that these biases arise from interacting causes rather than a single removable linear feature. For model-style sensitivity, one plausible source is reward overoptimization on imperfect signals: if the RM training data (or of the pretrained base model from which the RM is derived) overrepresents particular model families or stylistic conventions, the RM may learn to reward stylistic signatures independent of underlying answer quality. For sycophancy, prior work has linked RLHF to biased user-agreement behavior (Sharma et al., 2023), and our findings are consistent with that account. However, sycophancy is not a unitary phenomenon, and reflects an interaction between factors like annotation (or LLM-judge) bias, sensitivity to confidence cues, and the model’s own internal epistemic uncertainty. This perspective is also consistent with our empirical finding that sycophancy and model-style sensitivity are substantially more resistant to simple linear interventions than lower-complexity biases such as length, uncertainty, and position. Fixing these biases is a much more complex research problem that will require (intuitively) larger efforts in evaluation and data-cleaning methods alone.
>
> Sharma, M., et al. Towards Understanding Sycophancy in Language Models. ICLR 2024.
>
> We thank the reviewer for their feedback which allowed us to further strengthen and support our initial findings with these additional experiments.

---

> > ### Author Rebuttal · Reviewer_DDEb · 2026-04-03
> >
> > I appreciate the authors' detailed response to my questions. I will keep the positive rating.

---

> > > ### Author Response · Authors · 2026-04-04
> > >
> > > We sincerely thank Reviewer DDEb for the constructive engagement and for confirming that both concerns have been fully resolved.
> > >
> > > We note that the original score of 4 cited "limited evaluation" as the primary factor limiting impact. With the additional BoN/AlpacaEval experiments now demonstrating that our length probe generalizes to open-ended instruction-following and materially affects downstream selection under optimization (alongside our existing OOD results on RewardBench-2), we believe the evaluation is no longer limited in the way originally described.
> > >
> > > Beyond the debiasing method itself, we would like to highlight that the paper makes empirical contributions that stand independently:
> > > * the discovery of position bias and model-style reward sensitivity in RMs,
> > > * the finding that SOTA models trained to correct length bias have overcorrected in the opposite direction,
> > > * and a complexity taxonomy that provides actionable guidance on when linear interventions can and cannot succeed. We believe these findings are likely to inform both RM development and future debiasing research.
> > >
> > > Given that the stated basis for the score has been addressed, we would respectfully ask the reviewer to consider whether the current rating still reflects their updated assessment of the work. If there are aspects beyond the two originally raised concerns that the reviewer feels still limit the contribution, we would be happy to discuss them further.

---

### Official Review · Reviewer_nAa9 · 2026-03-13

**Soundness:** 3
**Presentation:** 3
**Significance:** 2
**Originality:** 2
**Overall Recommendation:** 4
**Confidence:** 4

**Summary:**

Reward models used in RLHF contain biases that language models learn to exploit. The paper measures five bias types across five RMs (length, uncertainty, position, sycophancy, model-style sensitivity) and proposes mechanistic reward shaping: find a bias direction in activation space via difference-of-means, then null-project it out at inference. No retraining required. The method works for simple linear biases and fails for complex entangled ones. Two novel biases are identified: position bias in RMs and model-style reward sensitivity

**Compliance With Llm Reviewing Policy:**

Affirmed.

**Final Justification:**

I'm adjusting my score to reflect the fact that my concerns have been adequately addressed.

**Key Questions For Authors:**

- Verbose responses are generated by rewriting concise ones with the same model, introducing confounds beyond length. Does the length probe generalize to verbose responses from a different model or human-written verbose responses?
- Could the authors provide even a small-scale experiment showing that RLHF or best-of-N sampling with the debiased RM produces better outputs? Without this the core motivation is not empirically validated.
- How was the 5 percentage point non-inferiority margin chosen, and does it reflect a principled tolerance for real RLHF settings?
- For model-style sensitivity, do high-correlation models correspond to models heavily represented in the RM's training data?

**Limitations:**

The limitations section acknowledges that length bias is only studied on math and that open-ended tasks are not covered, but does not discuss the three more fundamental issues: the synthetic nature of the controlled experiments and whether the probes capture the intended bias rather than confounds, the absence of any downstream policy evaluation, and the unjustified non-inferiority threshold.

**Strengths And Weaknesses:**

Strengths
- The low/high complexity bias taxonomy is a useful organizing contribution
- Model-style sensitivity is a novel and practically concerning finding: RMs reward outputs that resemble specific model families independent of content quality
- The method is data-efficient, model-internal, and requires no retraining
- The paper is honest about the limits of linear interventions, showing the sycophancy probe cannot reduce regressive agreement without hurting progressive agreement

Weaknesses:
- The controlled experiments use synthetic proxies that may not reflect real bias manifestations. Verbose responses are generated by prompting the same model to rewrite concise ones, introducing stylistic confounds beyond length. Uncertainty bias is tested by prepending a fixed hedging phrase. These are clean but narrow operationalizations.
- The paper never closes the motivating loop. The core claim is that RM biases cause bad downstream behavior, but no language model is ever trained with the debiased RM to verify this. The evaluation stops at bias reduction in the RM and approximate RewardBench-2 preservation, which are necessary but not sufficient
- The non-inferiority threshold on RewardBench-2 is 5 percentage points, which is generous and not justified.
Model-style sensitivity is documented but not mitigated. Mean effect sizes are weak (|ρ| ≈ 0.1), though individual model correlations reach 0.2-0.4.
- All evaluation is on verifiable multiple-choice tasks. How biases manifest in open-ended generation, the actual RLHF setting, is never tested.

---

> ### Author Rebuttal · Authors · 2026-03-31
>
> We thank Reviewer nAa9 for the detailed and constructive review, and for recognizing the value of the complexity taxonomy, model-style sensitivity finding, and the paper's transparency about limitations of linear interventions.
>
> **Does the length probe generalize?**
>
> While further tests would strengthen this claim, our evidence extends beyond the synthetic rewrite setting. In Appendix D.4, the length probe trained on GSM8K transfers to RewardBench-2, where it changes reward-length correlations in the expected direction across models, suggesting that it captures a transferable length-related feature rather than only artifacts of the synthetic construction. Additionally, our new BoN experiments on AlpacaEval (see below) provide further evidence. When we sample completions from Llama-3.2-1B-Instruct, the probe successfully reduces length exploitation in the BoN setting. Prior work (Kramar et al., 2026) shows that production probes can be fragile under distribution shift, making OOD evaluation especially important in our setting. Our RewardBench-2 transfer results therefore provide stronger evidence that the length probe captures a transferable feature rather than merely artifacts of the synthetic setup. We will add more clarifications and nuances to the discussion in the revision.
>
> Kramar et al. Building Production-Ready Probes for Gemini. arXiv:2601.11516 2026.
>
> **Synthetic Data and Downstream Policy Evaluation**
>
> To address real-world applicability, we evaluate reward models under Best-of-N (BoN) selection on AlpacaEval prompts, which directly probes behavior under optimization pressure. As N increases, baseline reward models amplify their inherent verbosity biases (DeBERTa longer, Allen shorter), while our debiased models attenuate or reverse these trends in agreement with our base paper findings. At N=32, the DeBERTa baseline produces substantially longer outputs than its debiased counterpart ($\sim$10 words), while the debiased Allen RM produces substantially longer outputs than baseline ($\sim$24 words), correcting each model’s bias direction. Under a GPT-5 AlpacaEval judge, this translates to downstream outcomes: the debiased DeBERTa RM achieves a higher win rate (17.19% vs 13.28%), while the debiased Allen RM achieves a comparable win rate (2.73% vs 3.71%). This demonstrates that debiasing materially affects selection.
>
> We also trained Llama-3.2-1B PPO policies (512 steps) with baseline and debiased RMs. For Allen, whose base RM penalizes verbosity (Section 4.1), the debiased policy produced significantly longer responses (+2.14 words, p=0.03) and was preferred by a GPT-5 judge in 52.3% of non-ties, consistent with end-to-end correction of overcorrection bias. For DeBERTa, differences from this small-scale policy optimization were not significant in either length or judge preference. We note that prior work demonstrating reward hacking via PPO relies on intentionally misspecified RMs with large bias effects, while our intervention targets subtle biases in production-grade RMs.
>
> **Non-inferiority threshold choice**
>
> The 5% margin reflects the statistical power available given small sample sizes of RewardBench-2 accuracy scores. Our observed accuracy differences are well within this margin (Table 2), so the threshold is conservative relative to the actual effect. While unintuitive for commonly saturated LLM benchmarks, the current leaderboard has a 5% difference between the top 5 SOTA models, and the top 10 models span 8%, justifying our  margin.
>
> **Multiple-Choice and free-form evaluation**
>
> For the concern that our evaluation is limited to multiple-choice settings, we agree that open-ended generation is the most important downstream regime. However we note that our position-bias analysis includes a quasi free-form setting in Appendix D.3, and the model-style sensitivity analysis is conducted on open-ended generations from Tulu-3/WildChat-derived data rather than MCQA alone. In addition, our new Best-of-N experiments on AlpacaEval directly evaluate optimization over free-form completions and show that debiasing meaningfully changes selection behavior in this setting.
>
> **Model-style sensitivity in open source datasets**
>
> Indeed, the tested model families are present in widely used open-source preference datasets, including LMArena’s Preference Proxy Evaluation, LMSYS Chatbot Arena and LMSYS-Chat-1M, and AllenAI’s Tulu 2.5 Preference Data. Thus, we argue that reward models reward features associated with recognizable model families that are present in these datasets or have been used to create synthetic (e.g., negative) samples. It is also reasonable to expect that the same effect extends to OpenAI- and Anthropic-family outputs, since such models are also present in widely used preference data, although we cannot test this without logit access. We will revise the paper to make these connections explicit.

---

> > ### Author Rebuttal · Reviewer_nAa9 · 2026-04-04
> >
> > The new BoN and PPO experiments meaningfully close the gap between the paper's motivation and its evidence. However, the PPO results only show significant effects for one of two tested RMs. I also note that while OOD transfer is demonstrated, the probe is still trained on synthetic rewrites, so it remains unclear whether it captures length specifically or also encodes stylistic artifacts of the rewriting process.

---

> > > ### Author Response · Authors · 2026-04-04
> > >
> > > We appreciate the reviewer's continued engagement and the acknowledgment that the experiments shared partially address the concerns. We want to provide an addition experiment and address the remaining probe-confound question directly.
> > >
> > > **Additional PPO results.** To address concerns about the initial PPO setup, we reran experiments at larger scale (4× optimization steps) and evaluated on 10,000 held-out HH prompts. Under this setting, the DeBERTa-debiased policy produces shorter responses than the baseline (−1.13 words on average; p=0.0023), consistent with reducing the learned length bias. While response length exhibits high variance across prompts in both conditions for the DeBERTa model, the mean shift is statistically significant.
> > >
> > > **The raised style-confound hypothesis** would require a single confounding factor to simultaneously account for three independent observations across multiple independent tests. If the length probe were encoding "Llama-3.1-8B rewriting style" rather than length, it would need to simultaneously explain three observations:
> > > * A: Opposite directional effects across RMs.
> > > * B: Transfer to a different generator under BoN.
> > > * C: Systematic effects on RewardBench-2.
> > >
> > > We provide evidence for all three:
> > > (On A) The probe reduces DeBERTa's reward-length Spearman correlation from 0.611 to 0.067, while slightly increasing correlation for newer RMs trained to penalize verbosity (e.g., Skywork-Llama from 0.267 to 0.305). A single stylistic confound would need to be rewarded by DeBERTa and penalized by SOTA models and this would need to align perfectly with each RM's independently measured length-bias direction. Thus, length is the parsimonious explanation.
> > >
> > > (On B) Our BoN experiments on AlpacaEval show that debiasing holds up under optimization pressure. As N increases, the debiased RMs attenuate or reverse the verbosity trends of the baseline, with opposite directional effects for different RMs. A style confound would need to systematically track selection pressure in opposite directions for different RMs, which is better explained by length than by a fixed stylistic artifact. Additionally, the original GSM8K probe, constructed entirely from Llama-3.1-8B-Instruct rewrites, transfers to RewardBench-2, which draws from diverse model families and human sources, providing direct cross-generator evidence (elaborated in C).
> > >
> > > (On C) This benchmark draws from diverse model families and human sources. A style confound tied to one model's rewriting patterns should produce noisy or null effects here, not the systematic reward-length decorrelation we observe (Figures 4–8).
> > >
> > > We acknowledge that the ideal ablation of retraining probes from verbose rewrites by a different model would further strengthen the case, and we plan to include this in the revision. **However, we note that the confound hypothesis must simultaneously explain all three observations above, while the length interpretation explains them straightforwardly**.
> > >
> > > Stepping back, we want to highlight the overall evidence of impact of our submission further supported by the new experiments:
> > > * a novel bias taxonomy with practical implications,
> > > * two previously uncharacterized RM failure modes (position bias and model-style sensitivity),
> > > * a lightweight and extensible debiasing method that requires no retraining,
> > > * OOD transfer validation on RewardBench-2 without significant accuracy degradation,
> > > * and new downstream BoN and PPO experiments confirming that debiasing propagates to model selection and policy training.
> > >
> > > We believe this represents a substantive contribution to the community's understanding of reward model failure modes and their mitigation, and we respectfully ask the reviewer to reconsider whether the current score fully reflects the breadth and strength of this evidence.

---

### Official Review · Reviewer_9JU3 · 2026-03-13

**Soundness:** 3
**Presentation:** 3
**Significance:** 3
**Originality:** 3
**Overall Recommendation:** 4
**Confidence:** 4

**Summary:**

This paper explores the persistent challenges of reward hacking in RLHF. The authors systematically assess biases in five sota reward models, revealing issues like length bias, sycophancy, and overconfidence, which still affect performance despite prior work. The paper introduces a novel mechanistic reward shaping approach designed to address low-complexity biases without compromising reward model quality. Through empirical evaluation, the authors demonstrate that their method not only mitigates biases but also generalizes well across diverse benchmarks.

**Compliance With Llm Reviewing Policy:**

Affirmed.

**Final Justification:**

The author has addressed my concerns. I will maintain my positive score.

**Key Questions For Authors:**

N/A

**Limitations:**

yes

**Strengths And Weaknesses:**

**Strengths:**

The paper makes a contribution by proposing a novel mechanistic reward shaping approach that targets low-complexity biases in reward models. This method effectively reduces these biases without compromising the overall reward quality. Furthermore, the paper demonstrates the generalizability of the proposed intervention across multiple benchmarks, offering compelling empirical results that suggest it could be applied to a wide range of tasks beyond the tested models. This paper is important for both improving the fairness and robustness of reward models and advancing RLHF research by providing a practical solution to bias-related issues.

**Weaknesses:**

While the paper effectively addresses low-complexity biases, it could benefit from a deeper discussion of the scalability of this approach to more complex bias types, such as sycophancy and model-style sensitivity.
The paper focuses mainly on benchmark datasets, and further exploration of real-world settings could strengthen its findings.

---

> ### Author Rebuttal · Authors · 2026-03-31
>
> We thank Reviewer 9JU3 for highlighting the importance of extensibility across domains. Our method enables targeted interventions on model representations, rather than relying on observable textual features.
>
> **Addressing High-Complexity Biases**
>
> We agree that mitigation of high-complexity biases is an important open problem. However, we emphasize that identifying these biases as structurally distinct is itself a contribution, supported by the novel biases we identify. Prior work has focused on spurious correlations (usually one at a time) that lead to reward misgeneralization under optimization. We instead identify a distinct regime of high-complexity biases, where the bias is entangled with the reward signal in a non-linear way and cannot be cleanly isolated. For example, sycophancy is co-linear with useful agreement signals (Section 5.1), meaning it cannot be removed without degrading reward quality.
>
> To begin working toward triaging high-complexity issues more effectively, we conducted further experimentation of linear separability of activations using iterative nullspace projection (INLP; Ravfogel et al., 2020), which tests whether bias signals remain linearly decodable after removing the top direction. For low-complexity biases (length, uncertainty, calibration), we observe strong initial separability (AUCs up to 1.00), but after projecting out the top direction, residual separability collapses to near chance (AUC ≈ 0.50–0.57). For high complexity biases, we observe weaker and more diffuse separability. For instance, splitting text that is higher-likelihood in different language models (Qwen2.5-7B and Llama2-7B) using our tokenization-agnostic method from the manuscript, we find that separability is near chance (AUC 0.58) for the Skywork–llama8b RM tested. For sycophancy, we find that progressive and regressive sycophancy are not fully separable (AUC = 0.68), using the same model. Since we exclusively look at the final layer of each RM, non-separability likely means that the bias exists independently in many pathways, versus via a single pathway by its own right, which could be modified. This points to more comprehensive data or training interventions, rather than simple reward shaping.
>
> Ravfogel, Shauli, et al. "Null it out: Guarding protected attributes by iterative nullspace projection." Proceedings of the 58th annual meeting of the association for computational linguistics. 2020.
>
> **Exploration of Real-world Settings**
>
> To address real-world applicability, we evaluate reward models under Best-of-N (BoN) selection on AlpacaEval prompts, which directly probes behavior under optimization pressure. As N increases, baseline reward models amplify their inherent verbosity biases  (DeBERTa longer, Allen shorter), while our debiased models attenuate or reverse these trends in agreement with our base paper findings. At N=32, the DeBERTa baseline produces substantially longer outputs than its debiased counterpart ($\sim 10$ words), while the debiased Allen RM produces substantially longer outputs than baseline ($\sim 24$ words), correcting each model’s bias direction. Under a GPT-5 AlpacaEval judge, this translates to downstream outcomes: the debiased DeBERTa RM achieves a higher win rate (17.19% vs 13.28%), while the debiased Allen RM achieves a comparable win rate (2.73% vs 3.71%) despite removing its bias toward shorter responses. This demonstrates that reward model biases materially affect selection under optimization, and that our method mitigates these effects.
>
> We also trained Llama-3.2-1B PPO policies (512 steps) with baseline and debiased RMs, generating on 1.5k held-out prompts. For Allen, whose base RM penalizes verbosity by 25pp in controlled evaluation (Section 4.1), the debiased policy produced significantly longer responses (+2.14 words, p=0.03) and was preferred by a GPT-5 judge in 52.3% of non-ties, consistent with end-to-end correction of overcorrection bias. For DeBERTa, differences were not significant in either length (+0.42 words, p=0.77) or judge preference (49.7% non-tie win rate, p=0.82), consistent with the KL constraint limiting deviation from an already-verbose reference policy. We note that prior work demonstrating reward hacking via PPO relies on intentionally misspecified RMs with large bias magnitudes; our intervention targets subtle biases in production-grade RMs, where BoN provides a more sensitive evaluation by directly probing ranking behavior under increasing selection pressure. The PPO and BoN results support our initial findings and we will add them to the final version of the paper.
>
> We thank the reviewer for their feedback which allowed us to further strengthen and support our initial findings with these additional experiments.

---

> > ### Author Rebuttal · Reviewer_9JU3 · 2026-04-04
> >
> > Thank you for your response. I have no additional questions about this submission. I will maintain my positive score.

---

> > > ### Author Response · Authors · 2026-04-04
> > >
> > > We thank Reviewer 9JU3 for confirming that our rebuttal fully addressed their concerns. We appreciate the positive assessment and would welcome the opportunity to engage further on the technical substance. We note that several aspects of the paper were not discussed in the review, and we would greatly value the reviewer's perspective on them. In particular:
> > >
> > > * We identify two previously uncharacterized biases in reward models (position bias and model-style reward sensitivity). The latter has significant implications for how RLHF training data is curated and how RMs are paired with policies. We'd welcome the reviewer's thoughts on the implications of these findings.
> > > * We find that state-of-the-art RMs trained to eliminate length bias have introduced a reverse length bias, preferring concise incorrect answers over verbose correct ones. We believe this is a practically important finding for RM developers. We are curious whether this changes the reviewer's assessment of the paper's impact.
> > > * Our rebuttal provided new INLP evidence showing that low-complexity biases collapse to near-chance separability after removing a single direction, while high-complexity biases remain diffusely encoded. We view this as a diagnostic framework the community can use to triage new biases, which is particularly impactful, given that most prior works focus only on one bias at a time. Does the reviewer see this as an independent contribution?
> > >
> > > We believe these contributions, taken together with the mechanistic reward shaping method, the extensive empirical evaluation across five RMs and four benchmarks, and new BoN/PPO generalization results, may warrant reconsideration of the score.
> > >
> > > We would equally welcome any specific technical reservations that informed the current rating, as the initial review did not include key questions, and we want to ensure we have the opportunity to address any remaining concerns.

---

### Decision · Program_Chairs · 2026-04-30

**Decision:**

Accept (regular)

**Comment:**

The paper considers the problem of bias in reward models. The authors run a series of experiments to uncover a combination of known biases in reward models as well as some new biases. Reviewers appreciated the importance of the problem considered and the intervention approach proposed. There were concerns about the data generation method for contrastive pairs and the evaluation procedure used. These concerns were partially addressed by new experiments, although some hesitation remains.

Overall, this is a paper where reviewers saw value in the contribution and the potential significance of the results. However, there is still room for improvement in the strength of the arguments and the comprehensiveness of the evaluation. This paper is on the borderline for acceptance.